# S-ATM: Self-Boosting Visual Reasoning via Adaptive Token Merging

## Abstract

VLMs, often adapted from LLMs, tend to show degraded reasoning capabilities when visual inputs are introduced. To address this issue, we propose S-ATM, a training-free decoding strategy that enhances visual reasoning without relying on external priors. For each input, two parallel pathways are constructed: one using the original image–text input and the other using a self-generated caption–text input. Their decoding distributions are adaptively merged at each step, with the merging weight guided by the model's attention to visual tokens. A momentum-based smoothing mechanism further stabilizes this merging over time. We conduct comprehensive experiments on diverse visual reasoning benchmarks to demonstrate the effectiveness of S-ATM. Further analysis shows that S-ATM primarily activates at high-entropy forking tokens, which often correspond to reasoning transitions, and that momentum smoothing reduces decoding instability and maintains reasoning coherence. These findings underscore the role of token-level dynamics in supporting long-chain reasoning in VLMs and further clarify how S-ATM works.

## 1 Introduction

Current mainstream vision–language models (VLMs) (Yao et al., 2024; Bai et al., 2025; Kimi Team, 2025; MiMo Team, 2025) extend large language models (LLMs) (Team, 2023; Dubey et al., 2024; Yang et al., 2025) with external visual encoders (Radford et al., 2021b; Zhai et al., 2023; Liu et al., 2025b), a paradigm first introduced by the LLaVA series (Liu et al., 2023; 2024). This design enables LLMs to process images and videos alongside text and has proven to be one of the most effective ways to leverage the strong reasoning ability of LLMs for visual tasks (Li et al., 2024; Chen et al., 2025b).

However, VLMs often fail to fully preserve the reasoning capacity of their LLM backbones (Zhang et al., 2024b; Qiao et al., 2024). A possible reason lies in *attention allocation*: visual tokens frequently attract non-negligible attention mass even in parts of a reasoning process where visual input is not directly needed (Zhang et al., 2025). For example, when solving a math problem with a diagram, the model must alternate between extracting visual details and performing symbolic reasoning. Continuous attention to visual tokens can break the reasoning flow and lower performance.

This issue can be mitigated through both training-based and training-free angles. Training-based methods (Zhou et al., 2025; Ding & Zhang, 2025) include supervised fine-tuning with curated reasoning traces (Chen et al., 2025a; Wang et al., 2025a) and reinforcement learning with verifiable rewards (Meng et al., 2025; Liu et al., 2025a; Wu et al., 2025), which have shown effectiveness but demand extensive computation and costly datasets. Training-free approaches offer greater flexibility, yet they remain relatively scarce. The most notable direction so far is model merging with external math LLMs (Chen et al., 2025b), which enhances reasoning but introduces additional dependencies and does not directly target the underlying issue of attention misallocation.

In this work, we propose S-ATM, a training-free decoding strategy that mitigates reasoning degradation in VLMs without extra supervision or reliance on external models. S-ATM constructs two decoding pathways: a visual pathway using the original image–text input, and a textual pathway that replaces the image with a self-generated caption. These two pathways are decoded in parallel, and their token distributions are merged at each decoding step using a weight that is adaptively adjusted based on the model's attention to visual tokens. A momentum-based smoothing mechanism further stabilizes the merging process over time. This lightweight design aims to improve reasoning capability of models while retaining essential visual grounding. A simplified overview is shown in Figure 1.

Figure 1: The overall pipeline of our proposed `S-ATM`, illustrating the caption generation and attention-driven token merging process at a single timestep. The image shows the target token, which refers to the output of `S-ATM`. It also displays the generated results from the Visual Pathway, Textual Pathway, and `S-ATM`.

We evaluate `S-ATM` across a wide range of model sizes, model families, and diverse visual reasoning benchmarks. Empirical results generally show consistent improvements, and performance on MathVerse especially stands out due to its emphasis on complex symbolic reasoning. To further assess the contribution of the textual pathway, we conduct a play-and-plug study where an external LLM is used to decode the caption–text input in place of the original VLM. This setup yields additional gains, indicating that the textual pathway effectively captures and supports multi-step reasoning.

Beyond empirical performance, we analyze how `S-ATM` shapes the decoding process to uncover its underlying mechanism. Our findings reveal two key insights:

**I. Merging activates at critical decision points.** `S-ATM` primarily intervenes at high-entropy forking tokens—positions where the visual and textual pathways produce divergent predictions. These tokens often mark reasoning transitions, such as sentence boundaries or operation-inducing verbs. Disabling `S-ATM` at these points notably reduces performance, underscoring their importance in guiding reasoning flow.

**II. Stability improves reasoning coherence.** Frequent pathway switching fragments the reasoning chain. To address this, `S-ATM` applies momentum-based smoothing to stabilize merging weights over time. We show that more appropriate switch frequency strongly correlates with higher accuracy, and proper tuning of hyperparameters consistently enhances decoding consistency.

Overall, we summarize our contributions as follows: ① We propose `S-ATM`, a training-free decoding strategy that effectively boosts visual reasoning by introducing a self-captioned textual pathway, and integrating attention-driven token merging with momentum-based smoothing. ② We conduct extensive experiments, where models could achieve over $2\%$ improvement in vision reasoning tasks without external assistance. In plug-and-play test, `S-ATM` successfully integrates the reasoning capabilities from other LLMs, achieving a remarkable $4.4\%$ improvement. These results validate the effectiveness and strong extensibility of `S-ATM`. ③ We provide in-depth analysis showing that `S-ATM` primarily acts on high-entropy forking tokens, and that decoding stability, achieved through momentum, is essential for maintaining coherent reasoning behavior.

## 2 BACKGROUND

**VLM decoding.** Given an image–text input pair $(\mathbf{I}, \mathbf{q})$, a vision–language model (VLM) with parameters $\theta_{\text{vlm}}$ generates an output sequence $\mathbf{y} = (y_1, \ldots, y_T)$. The distribution factorizes as

$$p_{\theta_{\text{vlm}}}(\mathbf{y} \mid \mathbf{I}, \mathbf{q}) = \prod_{t=1}^{T} p_{\theta_{\text{vlm}}}(y_t \mid \mathbf{y}_{<t}, \mathbf{I}, \mathbf{q}), \tag{1}$$

with the prefix $\mathbf{y}_{<t} = (y_1, \ldots, y_{t-1})$. At decoding step $t$, the model produces a logit vector $\mathbf{z}_t = p_{\theta_{\text{vlm}}}(\mathbf{y}_{<t}, \mathbf{I}, \mathbf{q})$, which defines a categorical distribution $\text{softmax}(\mathbf{z}_t)$ over the vocabulary; the next token $y_t$ is then sampled accordingly. Hence, the overall sequence distribution is induced by the collection of logit distributions $\{\text{softmax}(\mathbf{z}_t)\}_{t=1}^{T}$.

**Attention in VLMs.** At decoding step $t$, the model attends to a context $\mathcal{U}_t = [\mathcal{V}; \mathcal{Q}; \mathcal{Y}_{<t}]$, where $\mathcal{V}$ are visual tokens from the image, $\mathcal{Q}$ are text tokens from the query, and $\mathcal{Y}_{<t}$ are previously generated tokens. For each token $u_m \in \mathcal{U}_t$, the attention weight is $\alpha_{t,m} = \exp(s_{t,m})/\sum_l \exp(s_{t,l})$, with $s_{t,m}$ denoting its similarity score. The sum of attention weights over each group is defined as $r_t^{\text{vis}}$ for tokens in $\mathcal{V}$, $r_t^{\text{txt}}$ for tokens in $\mathcal{Q}$, and $r_t^{\text{gen}}$ for tokens in $\mathcal{Y}_{<t}$, which satisfy $r_t^{\text{vis}} + r_t^{\text{txt}} + r_t^{\text{gen}} = 1$.

**Challenge: on the degradation of reasoning in VLMs with visual information.** VLMs, typically extended from LLMs, are trained on large-scale vision–language corpora to align visual features to text. However, this process also introduces a structural bias: the attention distribution $(r_t^{\text{vis}}, r_t^{\text{txt}}, r_t^{\text{gen}})$ almost always assigns a non-negligible portion to visual tokens, even in decoding steps where image information is not directly required. This persistent presence of $r_t^{\text{vis}}$ diverts capacity from textual and contextual attention, which can disrupt multi-step reasoning and reduce continuity. A typical case is shown in Appendix A.4.

While additional training on curated chain-of-thought data may mitigate this issue, such resources are scarce and costly. Our method tackles the problem from an orthogonal perspective: *it introduces an auxiliary distribution derived from self-generated captions, adaptively merged with the original decoding distribution*. This integration aims to rebalance token-level dynamics and enables VLMs to exploit the reasoning capabilities of their LLM backbones without training.

## 3  S-ATM: A SELF-BOOSTING FRAMEWORK FOR VISUAL REASONING

To help VLMs self-boost the reasoning ability in their LLM backbones, we introduce S-ATM, a decoding strategy that runs two pathways in parallel: the original image–text input (visual pathway) and a self-generated caption–text input (textual pathway). The latter establishes a text-only route for VLMs, allowing it to focus solely on language by removing the distraction of image information. Specific designed S-ATM enables us to combine the strengths of both pathways. Their outputs are merged adaptively, as shown in Fig. 1.

### 3.1  CAPTION GENERATION

To create the two pathways (visual and textual), we first perform a text-level augmentation. We use VLM to generate a comprehensive, concise, and accurate caption $\mathbf{C}$ for the image. The detailed prompt are presented in Appendix A.2.

### 3.2  ADAPTIVE TOKEN MERGING

With the generated caption, we initialize two different inputs for the two pathways. The first vision pathway takes $\mathbf{I}$ and $\mathbf{q}$ as input, while the second textual pathway takes the $\mathbf{C}$ and $\mathbf{q}$ as input. This design allows VLMs to fully concentrate on the text in the textual pathway, free from interference by image tokens. Both pathways perform decoding simultaneously. At time step $t$, they generate their respective logits $\mathbf{z}_{\text{vis}}$ and $\mathbf{z}_{\text{res}}$:

$$\mathbf{z}_{\text{vis}} = p_{\theta_{\text{vlm}}}(\mathbf{y}_{<\mathbf{t}}, \mathbf{I}, \mathbf{q}), \quad \mathbf{z}_{\text{res}} = p_{\theta_{\text{vlm}}}(\mathbf{y}_{<\mathbf{t}}, \mathbf{C}, \mathbf{q}) \tag{2}$$

Subsequently, using the attention matrix, we can merge these logits. To compute the image attention ratio, we first extract the attention weights of a specific layer of $\theta_{\text{vlm}}$ at $t$. Let $A \in \mathbb{R}^{H \times L}$ denote the attention matrix, $L$ denote the sequence length at $t$, $L_{\text{img}}$ denote the number of image tokens and $H$ is the number of attention heads. The image attention ratio $\alpha_{\text{att}}$ is calculated as follows:

$$A_{\text{img}}^{\text{agg}} = \sum_{h=1}^{H} \frac{1}{L_{\text{img}}} \sum_{i=\text{img}_{\text{begin}}}^{\text{img}_{\text{end}}} \mathcal{A}[h, i], \qquad A_{\text{all}}^{\text{agg}} = \sum_{h=1}^{H} \frac{1}{L} \sum_{i=1}^{L} \mathcal{A}[h, i], \qquad \alpha_{\text{att}} = \frac{A_{\text{img}}^{\text{agg}}}{A_{\text{all}}^{\text{agg}}} \tag{3}$$

Next, we carefully map the $\alpha_{\text{att}}$ to a specific range for practical usage by defining a linear mapping function that transforms values less than 0.1 to $[-\epsilon, \epsilon]$, yielding the transformed value $\delta_{\mathbf{t}}$. The final adaptive weight is then computed as:

$$\alpha_t = \alpha_0 + \delta_{\mathbf{t}} \tag{4}$$

where $\epsilon$ and $\alpha_0$ are predefined hyperparameters. Using the computed adaptive weight, we can merge the logits from both pathways as follows:

$$\mathbf{z}_t = \alpha_t \cdot \mathbf{z}_{\text{vis}} + (1 - \alpha_t) \cdot \mathbf{z}_{\text{res}} \tag{5}$$

The final token $\mathbf{y}_t$ is sampled from $\mathbf{z}_t$ and added to both pathway's token sequence.

### 3.3 Momentum Smoothing

To stabilize the decoding process (i.e., avoid frequent switching between the twe pathways), inspired by the momentum concept in Adam (Kingma & Ba, 2017) algorithms, we introduce a hyperparameter $\gamma$ to ensure that $\alpha_t$ exhibits relatively more stable while decoding.

Specifically, we compute the smoothed alpha value and merged logits as:

$$\tilde{\alpha}_t = \gamma \cdot \alpha_t + (1 - \gamma) \cdot \tilde{\alpha}_{t-1}, \quad \mathbf{z}_t = \tilde{\alpha}_t \cdot \mathbf{z}_{\text{vis}} + (1 - \tilde{\alpha}_t) \cdot \mathbf{z}_{\text{res}} \tag{6}$$

To theoretically prove the efficiency of momentum smoothing in the decoding process, we consider a sequence of $N$ tokens generated by VLM and compute the expected times of visual/reasoning property switches between adjacent tokens as a measure of decoding process stability. The problem is simplified by modeling $\alpha_t$ at each timestep as $N$ independent and identically distributed random variables $\alpha_t \sim \mathbf{X}_t \overset{\text{i.i.d.}}{\sim} \mathcal{N}(0, \sigma^2)$. We use the expected times of sign flips between consecutive $\alpha_t$ as a measure of decoding stability. The smoothed sequence with momentum is defined as:

$$\mathbf{S}_1 = \mathbf{X}_1, \mathbf{S}_2 = \gamma \mathbf{X}_2 + (1 - \gamma)\mathbf{S}_1, ......, \mathbf{S}_i = \gamma \mathbf{X}_t + (1 - \gamma)\mathbf{S}_{t-1}, \quad \gamma \in (0, 1), t \in (1, N). \tag{7}$$

The sign flipping probabilities between two consecutive timestep are given by:

$$P_0 = \tfrac{1}{2} \quad (\gamma = 1) \tag{8}$$

$$P_{\text{mom},t} = \frac{1}{\pi} \arccos\left(\rho_{\mathbf{S}_t, \mathbf{S}_{t-1}}\right) = \frac{1}{\pi} \arccos\left(\frac{1 - \gamma}{\sqrt{\gamma^2 \frac{\sigma^2}{\text{Var}(\mathbf{S}_{t-1})} + (1 - \gamma)^2}}\right) \quad (\gamma \neq 1) \tag{9}$$

$\rho_{\mathbf{S}_t, \mathbf{S}_{t-1}}$ denotes the correlation coefficient between $\mathbf{S}_t$ and $\mathbf{S}_{t-1}$. Since $\arccos(\cdot)$ is strictly decreasing and $\rho_{\mathbf{S}_t, \mathbf{S}_{t-1}} > 0$, we have $\sum_{t=2}^{N} P_{\text{mom},t} < (N-1) \cdot P_0$. This shows that momentum smoothing could reduces reasoning mode switching frequency. Details are provided in Appendix A.3.

## 4 Experiments

This section primarily presents the experiment setup, main results, ablation study, and plug-and-play experiments. We discuss the performance exhibited by different models across various benchmarks through comprehensive experimental results, validating the effectiveness of `S-ATM`. Furthermore, we conduct an ablation study to demonstrate the necessity of each design choice in `S-ATM`. Through plug-and-play experiments, `S-ATM` surpasses previous model merging methods and exhibits strong flexibility as expected.

### 4.1 Experiment Setup

**Models.** We conduct experiments on `qwen2.5-vl-instruct` (Bai et al., 2025) and `internvl3.5` (Wang et al., 2025c) series models, with parameter size ranging from 3B to 14B.

Table 1: Performance of models with different sizes on four multi-modal reasoning benchmarks. The evaluation covers **MathVerse** (vision-dominant, including All, P-Geo. (plane geometry), S-Geo. (solid geometry), and Func. (function)), **MathVista** (including All, Gen. (general), and Math) and **MMStar** (All and Math). Regarding the methods, **CapIn** denotes adding captions into the prompt, **CapOut** indicates generating a caption first and then the answer, and **CapOnly** (Qiao et al., 2024) refers to providing only captions without the image, following a similar pattern constructed in previous works. Results in other benchmarks and models are shown in Appendix A.7 and Appendix A.8.

| Model | Method | MathVerse | | | | MathVista | | | MMStar | |
|---|---|---|---|---|---|---|---|---|---|---|
| | | All | P-Geo. | S-Geo. | Func. | All | Gen. | Math | All | Math |
| qwen2.5-vl-3b | Baseline | 34.9 | 37.5 | 18.5 | 39.0 | 62.3 | 69.1 | 57.4 | 51.4 | 63.6 |
| | CapIn | 31.9 ↓3.0 | 36.1 | 12.2 | 32.9 | 61.4 ↓0.9 | 66.1 | 57.4 | 50.4 ↓1.0 | 61.7 |
| | CapOut | 34.4 ↓0.5 | 37.7 | 17.7 | 36.5 | 57.4 ↓4.9 | 68.9 | 54.4 | 50.4 ↓1.0 | 50.1 |
| | CapOnly | 31.1 ↓3.8 | 35.3 | 11.8 | 32.1 | 47.0 ↓15.3 | 45.4 | 48.3 | 42.1 ↓9.3 | 58.0 |
| | S-ATM | **37.1** ↑2.2 | 38.6 | 20.2 | 44.7 | **64.5** ↑2.2 | 68.0 | 61.5 | **53.3** ↑1.9 | 63.6 |
| internvl3.5-8b | Baseline | 48.0 | 53.9 | 26.1 | 45.9 | 74.6 | 73.5 | 75.6 | 67.1 | 79.3 |
| | CapIn | 48.4 ↑0.4 | 54.5 | 23.5 | 47.2 | 74.0 ↓0.6 | 71.3 | 76.3 | 65.1 ↓2.0 | 78.8 |
| | CapOut | 45.3 ↓2.7 | 52.4 | 21.0 | 40.9 | 74.0 ↓0.6 | 72.2 | 75.6 | 66.1 ↓1.0 | 77.2 |
| | CapOnly | 46.1 ↓1.9 | 51.0 | 20.2 | 49.7 | 65.5 ↓9.1 | 63.5 | 67.2 | 61.9 ↓5.2 | 72.8 |
| | S-ATM | **50.0** ↑2.0 | 56.1 | 23.5 | 50.3 | **75.1** ↑0.5 | 77.0 | 72.8 | **67.2** ↑0.1 | 78.0 |
| internvl3.5-14b | Baseline | 53.4 | 61.4 | 25.2 | 49.1 | **75.5** | 74.4 | 76.5 | 67.5 | 82.6 |
| | CapIn | 51.1 ↓2.3 | 57.1 | 24.4 | 52.2 | 74.8 ↓0.7 | 71.5 | 77.6 | 66.6 ↓0.9 | 78.4 |
| | CapOut | 46.2 ↓7.2 | 50.5 | 24.4 | 49.1 | 74.0 ↓1.5 | 72.8 | 75.0 | 68.2 ↑0.7 | 76.4 |
| | CapOnly | 48.9 ↓4.5 | 54.9 | 22.7 | 49.1 | 66.8 ↓8.7 | 62.4 | 70.6 | 63.6 ↓3.9 | 77.6 |
| | S-ATM | **54.0** ↑0.6 | 59.8 | 31.1 | 52.2 | 75.2 ↓0.3 | 76.1 | 74.1 | **68.6** ↑1.1 | 79.6 |

Additionally, we incorporate qwen2.5-3b/7b-instruct (Yang et al., 2024) as the LLMs with stronger reasoning capabilities for plug-and-play extension experiments. For simplicity, Qwen series models in this section are referred to without the "-Instruct" suffix.

**Benchmarks.** For evaluation, we apply three benchmarks, MathVerse (Zhang et al., 2024a), Math-Vista (Lu et al., 2024) and MMStar (Chen et al., 2024). Among the three benchmarks, MathVerse contain only mathematical reasoning tasks and MMStar is mainly constructed by perceptual tasks. Meanwhile, MathVista is a diverse benchmark that includes both reasoning tasks and general VQA. Such benchmarks allow our results to reflect the effectiveness of S-ATM on different problems.

**Implementation details.** Experiments are conducted on NVIDIA RTX 3090 GPUs. During decoding, temperature is set to 0.1 for stability and reproducibility, with a maximum token length of 2048. For the adaptive merging process, we fix $\epsilon = 0.01$ and $\gamma = 0.9$. The parameter $\alpha_0$ ranges from 0.5 to 0.6, depending on the benchmark (lower for inference-heavy benchmarks). Specifically, we use the validation set to select the optimal $\alpha_0$ from the values [0.5, 0.51, 0.53, 0.55, 0.57], depending on the benchmark's reliance on visual information. The attention layer used to calculate $\alpha_{att}$ is the first layer. Details of the choice of the hyperparameters are shown in Appendix A.6.1 and Appendix A.6.2.

## 4.2 RESULTS

**Main results.** S-ATM provides consistent improvements across different model sizes and series on all benchmarks, as shown in Tab. 4. For instance, qwen2.5-vl-3b achieves improvements of 2.2%, 2.2%, and 1.9% on the three datasets respectively, while internvl3.5-8b shows a 2% increase on MathVerse. Larger model like internvl3.5-14b also achieves a 1.1% improvement on MMStar. We also implement several traditional caption augmentation methods (**CapIn**, **CapOut**) as baselines. Additionally, following the approach in Qiao et al. (2024), we implement **CapOnly** (which is just the setting of Textual Pathway) for comparison. The results demonstrate that S-ATM outperforms all caption augmentation baselines, proving that previous caption augmentation methods alone are insufficient for enhancing performance on intensive reasoning tasks. This further validates the necessity and effectiveness of S-ATM.

**Excellence in reasoning.** S-ATM demonstrates remarkable effectiveness and robustness in intensive reasoning tasks. As shown in Tab. 4, all three models perform well, with a notable improvement of around 5% in Func. of MathVerse. Its robustness in MathVerse is further evidenced by consistent performance across a wide parameter range ($\alpha_0 \in [0.5, 0.55]$), based on our experiments. Furthermore,

Table 2: Ablation study on four multi-modal reasoning benchmarks. The evaluation covers **MathVerse** (vision-dominant, including All, P-Geo. (plane geometry), S-Geo. (solid geometry), and Func. (function)), **MathVista** (including All, Gen. (general), and Math) and **MMStar** (All and Math). Regarding the methods, **ATM** indicates adaptive token merging (without momentum smoothing), We bold the best result for each benchmark, while the light-blue row highlights the best-performing task vectors on average.

| Model | Method | MathVerse | | | | MathVista | | | MMStar | |
|---|---|---|---|---|---|---|---|---|---|---|
| | | All | P-Geo. | S-Geo. | Func. | All | Gen. | Math | All | Math |
| qwen2.5-vl-3b | Baseline | 34.9 | 37.5 | 18.5 | 39.0 | 62.3 | **69.1** | 57.4 | 51.4 | 63.6 |
| | + ATM | 36.0 ↑1.1 | 38.2 | 16.8 | 43.4 | 63.6 ↑1.3 | 68.0 | 59.8 | 53.0 ↑1.6 | 66.8 |
| | S-ATM | **37.1** ↑2.2 | **38.6** | **20.2** | **44.7** | **64.5** ↑2.2 | 68.0 | **61.5** | **53.3** ↑1.9 | 63.6 |
| internvl3.5-8b | Baseline | 48.0 | 53.9 | 26.1 | 45.9 | 74.6 | 73.5 | 75.6 | 67.1 | **79.3** |
| | + ATM | 49.4 ↑1.4 | 54.5 | **26.1** | 50.3 | **75.2** ↑0.6 | 74.4 | **75.9** | 65.8 ↓1.3 | 78.4 |
| | S-ATM | **50.0** ↑2.0 | **56.1** | 23.5 | **50.3** | 75.1 ↑0.5 | **77.0** | 72.8 | **67.2** ↑0.1 | 78.0 |

some general VQA tasks within MathVista and MMStar also demand substantial reasoning. With S-ATM's self-boosted ability in reasoning, 8B and 14B models could achieve significant gains of 3.5% and 1.7% in Gen. of MathVista.

**Ablation study.** As shown in Tab. 2, adding adaptive token merging consistently improves performance across models and benchmarks. For example, qwen2.5-vl-3b shows a performance increase from 34.9% to 36.0% on MathVerse and from 51.4% to 53.0% on MMStar, demonstrating the effectiveness of the merging approach. Similar improvements are observed in other tasks and models. When momentum smoothing is incorporated into S-ATM, which enhances stability by smoothing the decoding process, further improvements are observed. For instance, internvl3.5-8b improves from 49.4% to 50.0% on MathVerse. These results show that ATM and momentum smoothing simultaneously improve models' vision reasoning ability by providing a more dynamic and controllable decoding process for them.

**Plug-and-play performance.** Our framework supports plug-and-play merging between different models without extra hyperparameter tuning. Experimental results in Tab. 3 validate this flexibility: a same-series merging of qwen2.5-vl models (3B+7B) brings a 3% improvement, while a cross-series combination with internvl3.5-8b achieves a 3.2% gain, demonstrating effective capability fusion. Furthermore, integrating with LLMs (qwen2.5-3b/7b) yields a significant improvement of 3.3% and 4.4%, respectively. The combination of qwen2.5-vl-3b and qwen2.5-3b even surpasses the previous notable math LLM and VLM merging method (Chen et al., 2025b) by 3.7%, strongly demonstrating the outstanding performance of our method in training-free vision rea-

Table 3: Plug-and-Play performance of different textual models on **MathVerse**, including All, P-Geo. (plane geometry), S-Geo. (solid geometry), and Func. (function). We also report the model merging results and Textual Pathway CapOnly (Since the LLM itself cannot process images, this is also the baseline performance of the qwen2.5-3b/7b and internvl3.5-8b on **MathVerse**) results for comparison.

| Visual | Textual | MathVerse | | | |
|---|---|---|---|---|---|
| | | All | P-Geo. | S-Geo. | Func. |
| qwen2.5-vl-3b | - | 34.9 | 37.5 | 18.5 | 39.0 |
| | qwen2.5-vl-3b | 37.1 ↑2.2 | 38.6 | 20.2 | 44.7 |
| | qwen2.5-3b | 38.2 ↑3.3 | 41.0 | 22.7 | 40.9 |
| | qwen2.5-vl-7b | 37.9 ↑3.0 | 41.2 | 19.3 | 41.5 |
| | internvl3.5-8b | 38.1 ↑3.2 | 39.8 | 21.8 | 44.7 |
| | qwen2.5-7b | 39.3 ↑4.4 | 43.7 | 20.2 | 39.6 |
| **Model Merging (Chen et al., 2025b)** | | | | | |
| qwen2.5-vl-3b | qwen2.5-3b | 34.5 | 36.5 | 20.2 | 39.0 |
| **Textual Pathway CapOnly (Qiao et al., 2024)** | | | | | |
| | qwen2.5-vl-3b | 31.1 | 35.3 | 11.8 | 32.1 |
| | qwen2.5-3b | 37.9 | 41.0 | 21.8 | 40.3 |
| | qwen2.5-vl-7b | 34.9 | 39.2 | 9.2 | 40.3 |
| | internvl3.5-8b | 39.2 | 44.1 | 18.5 | 39.0 |
| | qwen2.5-7b | 36.6 | 41.6 | 16.0 | 36.4 |

soning, where boosting performance is highly challenging. These results collectively demonstrate S-ATM's strong extensibility and practical plug-and-play ability.

**Computation & Time Cost** S-ATM introduces additional pathway with extra, but acceptable, costs. In this section, we present S-ATM's computation and time costs. First, the adaptation token merging actually takes very little time. On average, token merging accounts for only 0.43% of the total time per token, though the total number of tokens increased by x1.26. To enable a fair comparison, we added an experiment where the baseline runs two inferences and record the pass@2 accuracy. In

this setting, the baseline's FLOPs are the same as ours (and even exceed `S-ATM`), allowing for a more equitable comparison. As shown in Tab. 4, `S-ATM` still achieves better performance, further demonstrating the effectiveness of our method under stricter comparisons as a viable inference time scaling method.

During actual implementation, decoding can be parallelized on a single model by setting the batch size to 2 (one for the visual pathway and one for the textual pathway), and the additional memory overhead does not double (from 8 GiB to 13 GiB for a 3B model). Based on the above analysis, `S-ATM` can be understood as an parallel decoding strategy that appropriately increases the FLOPs (with two pathways) and the number of tokens (since the reasoning process typically generates more tokens) to achieve stronger visual reasoning ability.

# 5 ANALYSIS

In this section, we analyze how `S-ATM` shapes the decoding process in visual reasoning. We first describe its general behavior, showing how perception and reasoning are interleaved through two pathways. We then examine the mechanism of token-level merging, focusing on the role of forking tokens in combining $\mathbf{z}_{vis}$ and $\mathbf{z}_{res}$. Finally, we discuss the necessity of stability in decoding and show how momentum provides explicit control over pathway switching. All the experiments in this section are conducted using `qwen2.5-vl-3b-instruct`.

Table 4: Comparison between `S-ATM` and the baseline's pass@2 version on MathVerse. Using `qwen2.5-vl-3b`.

| Method | MathVerse | | | |
|---|---|---|---|---|
| | All | P-Geo. | S-Geo. | Func. |
| Baseline | 34.9 | 37.5 | 18.5 | 39.0 |
| Baseline (Pass@2) | 36.7 ↑1.8 | 39.2 | 17.6 | 42.8 |
| `S-ATM` | **37.1** ↑2.2 | **38.6** | **20.2** | **44.7** |

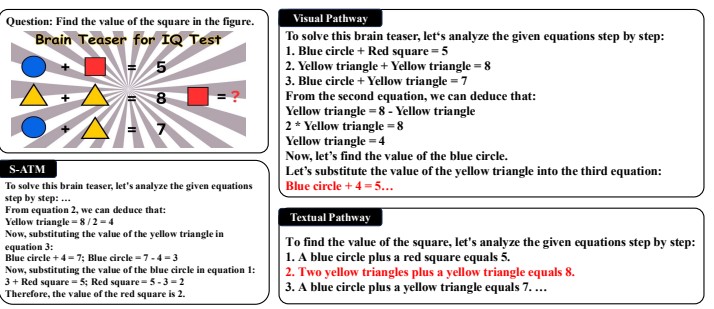

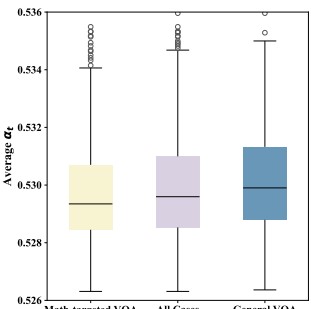

Figure 2: A typical case in visual reasoning. Comparison between the two pathways and `S-ATM`. Full version are shown in Appendix A.5

Figure 3: Average $\alpha_t$ of the cases in MathVista.

## 5.1 OVERVIEW: VISUAL REASONING TASKS INTERLEAVE PERCEPTION AND REASONING

**Task characteristics.** Visual reasoning tasks inherently require the interleaving of two abilities: perception of visual content and reasoning over contextual information. The visual pathway is effective at extracting perceptual details but tends to degrade as the context becomes longer, whereas the textual pathway provides more stable reasoning but often misses fine-grained visual cues. As illustrated in Fig. 2, neither pathway alone is sufficient, since perception and reasoning must interact throughout the task.

**Adaptive behavior of `S-ATM`.** `S-ATM` is designed to adapt to this interleaved requirement by dynamically adjusting the contributions of the two pathways during decoding. As shown in Fig. 3, analysis on 1,000 MathVista examples indicates that it assigns slightly more weight to the textual pathway in reasoning-intensive tasks (e.g., math problems), while maintaining stronger reliance on perception in more visually grounded cases (e.g., general problems).

## 5.2 ANALYSIS ON CAPTION QUALITY

In this section, we discuss the impact of caption quality on performance. First, since `S-ATM` is designed as self-boosting, the captions used in Tab. 4 are self-generated by the models. Here, we further conducted experiments comparing different quality captions' influence on the results, using `qwen2.5-vl-3b`. We experimented with self-generated captions and two high-quality captions: captions generated by `internvl3.5-8b` and MathVerse's text-dominant caption. As shown in Tab. 5, higher quality indeed helps the model pass more information through the textual pathway, enhancing the model's reasoning ability. Overall, we would like to emphasize that `S-ATM` already works well in the self-boosting setting without external information, and becomes even better when high-quality captions are available. This highlights the robustness of `S-ATM`.

Table 5: Comparison of `S-ATM`'s performance under different caption qualities. 8B's caption is generated by `internvl3.5-8b` and TD caption is MathVerse's official text dominant caption.

| Method | MathVerse | | | |
|---|---|---|---|---|
| | All | P-Geo. | S-Geo. | Func. |
| Baseline | 34.9 | 37.5 | 18.5 | 39.0 |
| S-ATM | 37.1 ↑2.2 | 38.6 | 20.2 | 44.7 |
| S-ATM + 8B's caption | 38.8 ↑3.9 | 41.4 | 21.8 | 43.4 |
| S-ATM + TD caption | 38.8 ↑3.9 | 42.5 | 21.6 | 39.6 |

### 5.3 FORKING TOKENS ARE THE KEY TO FUSE THE DECODING PATHS

The preceding subsection shows that the visual and textual pathways play complementary roles, with `S-ATM` dynamically adjusting their contributions according to task demands (Fig. 3). While this explains the overall balance between perception and reasoning, it does not reveal how merging occurs during decoding. Since the two pathways are fused token by token rather than after producing full sequences, it is necessary to examine decoding at the token level. A central question then arises: *at which tokens does `S-ATM` intervene, and how do these tokens shape the trajectory of reasoning?*

**Hypothesis: `S-ATM` acts on forking tokens.** We hypothesize that `S-ATM` primarily works on a specific class of tokens, which we refer to as *forking tokens*. Forking tokens are high-entropy tokens that often appear at sentence boundaries, transitional markers, or verbs that determine subsequent reasoning steps (e.g., "however," "so," "let us do"), a notion we adopt from prior work showing that high-entropy tokens frequently dictate reasoning direction (Wang et al., 2025b).

**Analysis through effective tokens.** To examine this hypothesis, we first define *effective tokens* as positions where the distribution $z_{vis}$ from visual pathway and the distribution $z_{res}$ from textual pathway yield different top-ranked predictions. Only at such positions can merging influence the output. As shown in Tab. 6, effective tokens exhibit substantially higher entropy (greater than 1.00) than the overall token population (less than 0.15), consistent with the defining properties of forking tokens. Moreover, the distributions of effective tokens and

Table 6: Entropy statistics of effective tokens across **MathVerse**, **MathVista**, and **MMStar**. To be specific, **All/Eff Ent.**: Average Entropy of all/effective tokens; **Eff./Fork**: The proportion of effective tokens among forking tokens; **Eff./All**: The proportion of effective tokens among all tokens; **Fork/Eff.**: The proportion of forking tokens among effective tokens; **Fork/All**: The proportion of forking tokens among all tokens; **MI**: The mutual information between the distribution of effective tokens and forking tokens.

| | MathVerse | MathVista | MMStar |
|---|---|---|---|
| All Ent. | 0.05 | 0.01 | 0.13 |
| Eff. Ent. | 1.04 | 1.14 | 1.08 |
| Eff./Fork (%) | 81.22 | 65.17 | 82.09 |
| Eff./All (%) | 3.00 | 3.00 | 8.50 |
| Fork/Eff. (%) | 100.00 | 100.00 | 100.00 |
| Fork/All (%) | 4.00 | 4.80 | 10.30 |
| MI | 0.80 | 0.65 | 0.78 |

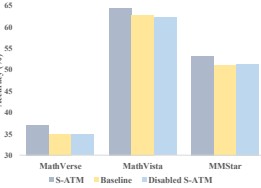

Figure 5: Comparison among disabled `S-ATM` at forking tokens, baseline, and the standard operation of `S-ATM`.

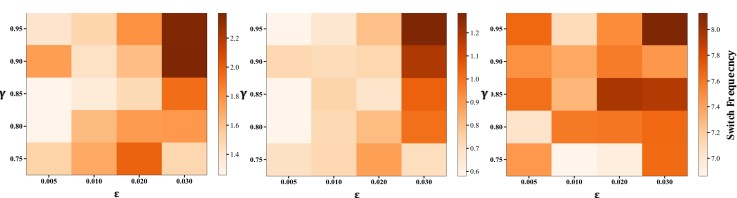

Figure 6: Heatmap of switch frequency with varying $\gamma$ and $\epsilon$. From left to right are the combination of all three benchmarks, MathVerse, and MMStar, each with 50 selected cases. Darker color means higher switch frequency.

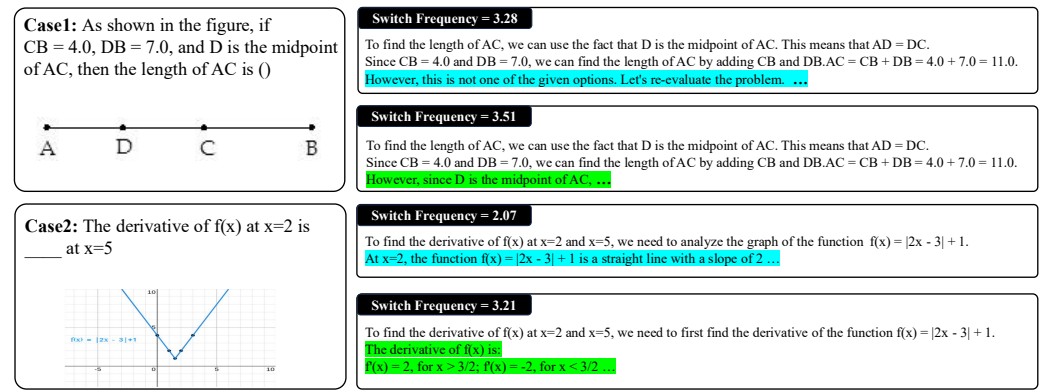

Figure 4: Two cases in visual reasoning. Comparison between relatively moderate and high switch frequency. Full version are shown in Appendix A.5.

forking tokens demonstrate strong statistical dependence, with mutual information values around 0.7 across benchmarks. This close alignment suggests that the effective tokens where S-ATM takes effect are essentially the forking tokens.

**Empirical evidence.** We further support our hypothesis with controlled experiments. Specifically, we disable S-ATM only at forking tokens, forcing the model to follow a single pathway at these high-entropy positions. As shown in Fig. 5, this modification leads to a clear drop in performance compared to the standard operation of S-ATM. These results indicate that the mechanism of S-ATM is indeed realized at forking tokens: by reshaping predictions at these critical junctures, we are able to dynamically steer reasoning trajectories toward more reliable outcomes without sacrificing perception.

**Case study.** We illustrate this effect with a qualitative example in Fig. 2. At the token "Now," the textual pathway $z_{res}$ correctly anticipates a "substituting" step and immediately leads the reasoning process of S-ATM to a faster direction, while the visual pathway $z_{vis}$ remains uncertain about what to do next. By merging the two, S-ATM selects the reasoning-consistent option. This example shows how forking tokens can steer the model to a more accurate reasoning path.

## 5.4 WHY WE NEED MOMENTUM — THE SIGNIFICANCE OF DECODING STABILITY

**Decoding stability as a requirement.** The stability of decoding is also crucial for VLMs, which is controlled by the smoothing process of momentum. To quantitatively evaluate the stability, for a token, whether it ultimately follows $z_{vis}$ or $z_{res}$ represents which pathway it follows at specific decoding step. If two consecutive tokens follow different pathways, we say that a switch occurs here. In a decoding sequence, the frequency that switch occurs is called switch frequency. We believe that the smaller the switch frequency, the longer the length of consecutive decoding sequences that stably follow a certain pathway, and thus the more stable the decoding process.

**Quantitative characterization.** To evaluate the smoothing effect of momentum, we vary $\gamma$ ($0.75 \sim 0.85$) and $\epsilon$ ($0.005 \sim 0.030$) and measure the switch frequency on 50-example subsets of each dataset. As shown in Fig. 6, increasing either $\gamma$ or $\epsilon$ consistently raises the switch frequency, from about $1.4\%$ to above $2.3\%$ in the three benchmarks—a relative increase of more than $1.5\times$. This demonstrates that momentum directly controls the stability of S-ATM.

We further examine accuracy under different switch frequencies (Fig. 7). Because frequency ranges vary across benchmarks, the $x$-axis is presented on a relative rather than absolute scale. Results support that only moderate values yield the best performance. The results reveal a clear trend: decoding stability, as reflected by lower switch frequency, is a key factor for achieving higher accuracy in visual reasoning tasks.

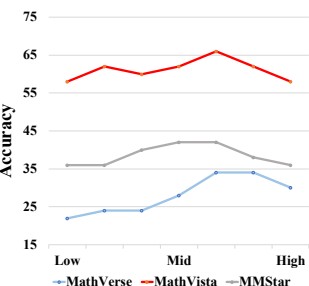

Figure 7: Accuracy versus switch frequency on all the benchmarks.

**Empirical evidence from case studies.** We highlight two representative cases in Fig. 4. In Case 1, excessive switching diverts the model at the green-marked sentence into the visual pathway, where it focuses on locating point D rather than continuing the reasoning chain. This prevents the correction step, just like the blue-marked position, that should have occurred. In Case 2, frequent back-and-forth switching at the green-marked sentence disrupts the construction of a perception sequence, and the model fails to correctly identify the slope of a line. Both cases demonstrate that when switch frequency is too high, perception and reasoning can't organized into effective continuous segments.

## 6 RELATED WORK

**VLMs.** Built upon large language models such as (Yang et al., 2025; Grattafiori et al., 2024), VLMs use an image encoder (CLIP (Radford et al., 2021a), InternViT, ViT (Dosovitskiy et al., 2021), etc.) and a projector (typically MLP) to integrate visual and textual information into a unified space for solving vision-language tasks. Recently, VLMs based on reinforcement learning or pretraining have begun to incorporate reasoning capabilities, enabling these models to handle more complex multimodal reasoning and achieving ever better performance on mathematical benchmarks ( (Zhang et al., 2024a; Lu et al., 2024; Wang et al., 2024)). Such benchmarks particularly examine the model's ability to coordinate between perception and reasoning, while also including the accuracy of each capability individually. This has made VLM reasoning & perception ability a more promising and worthy topic for discussion compared to perception capabilities.

**Visual reasoning in VLMs.** To achieve better performance on many reasoning-intensive circumstances, basic training-based approaches include (Zhou et al., 2025; Ding & Zhang, 2025). In addition, some works enhance reasoning ability by constructing high-quality large-scale datasets and verifiable rewards for training (Chen et al., 2025a; Wang et al., 2025a; Xu et al., 2024; Meng et al., 2025; Liu et al., 2025a; Wu et al., 2025). Beyond these training approaches, training-free methods are relatively rare. Among them, (Chen et al., 2025b) leverage external LLMs' textual ability and merge their parameter with VLMs, enabling VLMs to do reasoning better.

**Token merging.** Token merging has recently become a widely explored method to integrate the strengths of multiple models. For instance, ProxyThinker (Xiao et al., 2025) enables Large Language Models (LLMs) to function as VLMs by using small visual reasoners to modify specific tokens. To address perception hallucinations, methods such as (Wang et al., 2025d; Leng et al., 2024) employ contrastive decoding, using attention-related or image-augmentation methods to mitigate models' misunderstandings of images. Similarly, RITUAL (Woo et al., 2024) applies test-time image augmentation and parallel decoding to achieve the same goals. While these works primarily enhance models' perception quality, boosting VLMs' vision reasoning ability remains an underexplored area.

## 7 CONCLUSION

To enhance the vision reasoning ability of VLMs without training, we propose `S-ATM`, an adaptive token merging method that helps the model self-boost its reasoning capabilities within its LLM backbone. On various models and vision reasoning benchmarks, our method consistently achieves improvements, with a gain of up to $5.5\%$ on function reasoning tasks. In plug-and-play test, when combined with LLMs, `S-ATM` also achieve remarkable performance. These results validate the effectiveness and extensibility of our approach. In addition, we provide an in-depth analysis of the underlying mechanism of token merging and the impact of decoding stability, hoping our findings can offer valuable insights and open up new possibilities for VLM's token-level research.

## REPRODUCIBILITY STATEMENT

To ensure the reproducibility of our experiments, we provide a complete implementation of `S-ATM` on `qwen2.5-vl` series models for MathVerse in supplementary materials, along with detailed running instructions. This comprehensive guide and code demonstrate our methodology and hyperparameter selection, aiming to help users understand all the details in `S-ATM` and reproduce the same experimental results as ours.

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

## A  APPENDIX

### A.1  USE OF LLMs

Large language models were used to support writing (check grammar errors, improve the flow of writing, adjust content formatting) and debugging.

All core ideas, implementations, experiments and analysis were independently developed by the authors. No part of the scientific contribution was generated by an LLM.

### A.2  PROMPT FOR CAPTION GENERATION

Here, we present the prompt used for the first step: Caption Generation in `S-ATM`:

```
Generate an extremely detailed caption for this image, describing
every visible element with precision.  Include:  All objects
(e.g.  bottle, phone, animals, ...)  Exact text (e.g.  content,
font style, size, color, orientation,) Spatial layout (e.g.
distances between objects, depth ordering, angles) Lighting (e.g.
direction, intensity, shadow details, reflections) Subtle details
(e.g.  brand logos, wear patterns, texture specifics) or any other
important elements you think are important.  It doesn't have to
be long or complicated.  Be accurate and sufficient.  Don't add
information that isn't in the image.
```

### A.3  MATHEMATICAL PROOF OF MOMENTUM'S EFFECTIVENESS

We consider a simplified problem. Assume that at timestep $t$, the switching between visual/textual properties of adjacent tokens is determined by whether $\alpha_{t-1}$ and $\alpha_t$ relative to 0.5 have different magnitudes. If they differ, it indicates that the decoding property has switched at this point.

To further simplify the problem, we abstract each $\alpha_t$ as $\mathbf{X}_t \sim \mathcal{N}(0, \sigma^2)$, where the variables are mutually independent. We need to consider the expected times of sign flips between adjacent $\alpha$ in a long sequence of $N$ tokens. A higher expectation indicates more frequent switching of decoding properties and less stable reasoning, while a lower expectation indicates more stable reasoning.

Assume that $\alpha_t$ is independent and identically distributed as

$$\mathbf{X}_t \overset{\text{i.i.d.}}{\sim} \mathcal{N}(0, \sigma^2), \quad t \in (1, N)$$

**Without momentum.**  The probability that two consecutive tokens having different sign is

$$P_0 := \Pr\big(\operatorname{sign}(\mathbf{X}_t) \neq \operatorname{sign}(\mathbf{X}_{t-1})\big).$$

Since the variables are independent and symmetric around zero, we have $\Pr(\mathbf{X}_t > 0) = \frac{1}{2}$, hence

$$P_0 = 2\Pr(\mathbf{X}_t > 0, \mathbf{X}_{t-1} < 0) = 2 \cdot \tfrac{1}{2} \cdot \tfrac{1}{2} = \tfrac{1}{2}.$$

**With momentum.**  Introduce the smoothed sequence

$$\mathbf{S}_1 = \mathbf{X}_1, \mathbf{S}_2 = \gamma\mathbf{X}_2 + (1-\gamma)\mathbf{S}_1, ......, \mathbf{S}_i = \gamma\mathbf{X}_t + (1-\gamma)\mathbf{S}_{t-1}, \quad \gamma \in (0,1), \quad t \in (1, N).$$

Because $\mathbf{X}_t$ are Gaussian, the pair $(\mathbf{S}_t, \mathbf{S}_{t-1})$ is also Gaussian. We compute their covariance and variance:

$$\operatorname{Var}(\mathbf{S}_t) = \gamma^2\sigma^2 + (1-\gamma)^2 \operatorname{Var}(\mathbf{S}_{t-1})$$
$$\operatorname{Cov}(\mathbf{S}_t, \mathbf{S}_{t-1}) = (1-\gamma) \operatorname{Var}(\mathbf{S}_{t-1})$$

Thus the correlation coefficient is

$$\rho_{\mathbf{S}_t, \mathbf{S}_{t-1}} = \frac{\operatorname{Cov}(\mathbf{S}_t, \mathbf{S}_{t-1})}{\sqrt{\operatorname{Var}(\mathbf{S}_t) \operatorname{Var}(\mathbf{S}_{t-1})}} = \frac{(1-\gamma) \operatorname{Var}(\mathbf{S}_{t-1})}{\sqrt{(\gamma^2\sigma^2 + (1-\gamma)^2\operatorname{Var}(\mathbf{S}_{t-1})) \operatorname{Var}(\mathbf{S}_{t-1})}}$$

$$\rho_{\mathbf{S}_t, \mathbf{S}_{t-1}} = \frac{(1 - \gamma)}{\sqrt{\gamma^2 \frac{\sigma^2}{\mathrm{Var}(\mathbf{S}_{t-1})} + (1 - \gamma)^2}}$$

Note that $\rho_{\mathbf{S}_t, \mathbf{S}_{t-1}} > 0$ for any $\gamma \in (0, 1)$.

**Sign-flip probability.** For a zero-mean bivariate normal pair $(\mathbf{U}, \mathbf{V})$ with correlation coefficient $\rho$, the probability that their signs differ is given by

$$\Pr(\mathrm{sign}(\mathbf{U}) \neq \mathrm{sign}(\mathbf{V})) = \frac{1}{\pi} \arccos(\rho).$$

At timestep $t$, applying this to $(\mathbf{S}_t, \mathbf{S}_{t-1})$, we obtain

$$P_{\mathrm{mom},t} = \frac{1}{\pi} \arccos(\rho_{\mathbf{S}_t, \mathbf{S}_{t-1}}).$$

Since $\arccos(\cdot)$ is strictly decreasing and $\rho_{\mathbf{S}_t, \mathbf{S}_{t-1}} > 0$, we conclude

$$P_{\mathrm{mom},t} < \frac{1}{\pi} \arccos(0) = \tfrac{1}{2} = P_0.$$

Therefore, for a sequence of $N$ tokens, the expected number of sign flips with momentum is strictly less than without momentum:

$$\sum_{t=2}^{N} P_{\mathrm{mom},t} < (N - 1) \cdot P_0 = \frac{N - 1}{2} \tag{10}$$

**Conclusion.** Therefore, introducing momentum strictly reduces the expected number of sign flips between consecutive tokens.

### A.4 CASE STUDY FOR BACKGROUND

As shown in Fig. 8, this is a typical example where too much perception negatively impacts reasoning ability. In this sample, model's $\alpha_{att}$ is very high, significantly above the average value in this benchmark. This indicates that during the decoding process, the model overfocuses on the image, neglecting the requirements for the sample values to be considered in the task, ultimately leading to an error. In fact, this is a very simple task for an LLM backbone if it is given the image information, but it was done incorrectly due to the excessive influence of perception.

### A.5 CASE STUDY FOR ANALYSIS

As shown in Fig. 9 10 11, we provide the full versions of the cases presented in Section 5 of the main text.

### A.6 FURTHER ABLATION STUDY

#### A.6.1 ABLATION ON ATTENTION LAYERS

For the attention layer, we present a comparison of S-ATM's performance under different attention layers used in the computation of $\alpha_{att}$. As shown in Tab. 7, using shallower attention layers yields the best performance, which aligns with the common understanding in VLMs that shallower layer attention typically focuses on aligning low-level visual features with textual tokens (perception + grounding).

#### A.6.2 ABLATION ON $\alpha_0$

As shown in Tab. 8, we present a comparison of S-ATM's performance under different $\alpha_0$. The best-performing value, 0.51, is also the $\alpha_0$ we use on MathVerse.

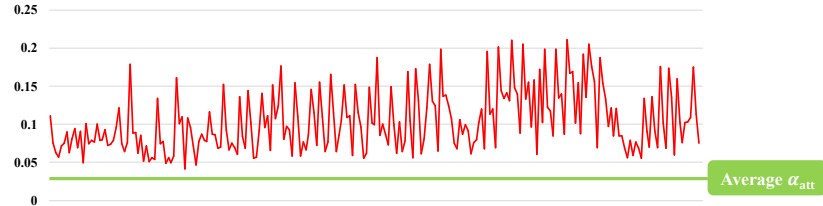

**Question:** The movie critic liked to count the number of actors in each movie he saw. How many movies had at least 30 actors but fewer than 47 actors?

**Ground Truth: 5**

Actors per movie

| Stem | Leaf |
|---|---|
| 2 | 2 3 3 3 9 |
| 3 | 6 9 |
| 4 | 3 3 5 |
| 5 | 4 4 5 |

**Generated Results**

Let's break down the information given in the stem and leaf plot: The stem represents the tens digit, and the leaf represents the ones digit. We are looking for movies with at least 30 actors but fewer than 47 actors. From the plot, we can see: For the stem 3, the leaves are 6 and 9, which represent 36 and 39. For the stem 4, the leaves are 3, 3, and 5, which represent 43, 43, and 45. For the stem 5, the leaves are 4 and 5, which represent 54 and 55. Now, let's count the movies that fall within the range of 30 to 46: 36, 39 (2 movies) 43, 43, 45 (3 movies) 54, 55 (2 movies) Adding these up, we get a total of 2 + 3 + 2 = 7 movies. Therefore, the answer is 7.

Figure 8: A typical case showing that excessive perception can impair reasoning in vision reasoning tasks. In addition, we provide the full decoding trajectory of $\alpha_{att}$ together with the average $\alpha_{att}$ on the benchmark for reference.

**Question: Find the value of the square in the figure.**

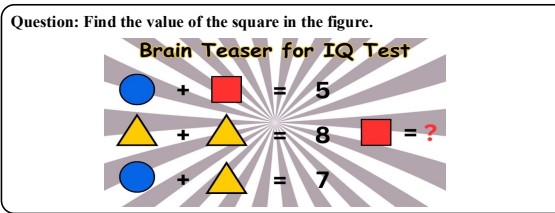

**S-ATM**

To solve this brain teaser, let's analyze the given equations step by step:
1.Blue circle + Red square = 5
2.Yellow triangle + Yellow triangle = 8
3.Blue circle + Yellow triangle = 7
From equation 2, we can deduce that:
Yellow triangle = 8 / 2 = 4
Now, substituting the value of the yellow triangle in equation 3:
Blue circle + 4 = 7; Blue circle = 7 - 4 = 3
Now, substituting the value of the blue circle in equation 1:
3 + Red square = 5; Red square = 5 - 3 = 2
Therefore, the value of the red square is 2.

**Visual Pathway**

To solve this brain teaser, let's analyze the given equations step by step:
1. Blue circle + Red square = 5
2. Yellow triangle + Yellow triangle = 8
3. Blue circle + Yellow triangle = 7
From the second equation, we can deduce that:
Yellow triangle = 8 - Yellow triangle
2 * Yellow triangle = 8; Yellow triangle = 4
Now, let's find the value of the blue circle.
Let's substitute the value of the yellow triangle into the third equation:
Blue circle + 4 = 5; Blue circle = 5 – 4; Blue circle = 1
Finally, let's substitute the values of the blue circle and yellow triangle into the third equation:
1 + 4 = 5; 5 = 5
Now, we can find the value of the red square:
Blue circle + Red square = 5; 1 + Red square = 5
Red square = 5 – 1; Red square = 4
Therefore, the value of the red square is 4.

**Textual Pathway**

To find the value of the square, let's analyze the given equations step by step:
1.A blue circle plus a red square equals 5.
2.Two yellow triangles plus a yellow triangle equals 8.
3.A blue circle plus a yellow triangle equals 7.
From the second equation, we can deduce that:
2 yellow triangles + yellow triangle = 8
This simplifies to:
3 yellow triangles = 8
So, one yellow triangle equals:
yellow triangle = 8 / 3
Now, using the first equation:
blue circle + red square = 5
We can substitute the value of the yellow triangle:
blue circle + red square = 5
blue circle + 8 / 3 = 5
To isolate the blue circle, we subtract 8 / 3 from both sides:
blue circle = 5 - 8 / 3
blue circle = 15 / 3 - 8 / 3
blue circle = 7 / 3
Now, using the third equation:
blue circle + yellow triangle = 7
We substitute the value of the blue circle:
7 / 3 + 8 / 3 = 7
15 / 3 = 7
5 = 7
This is a contradiction, which suggests that the values provided in the equations might not be consistent. However, if we assume the values are correct, we can conclude that the square must be the remaining part of the equation:
red square = 5 - blue circle
red square = 5 - 7 / 3
red square = 15 / 3 - 7 / 3
red square = 8 / 3
Therefore, the value of the square is:
8 / 3

Figure 9: From MathVista, $pidx = 98$. Including results generated by Visual Pathway, Textual Pathway and S-ATM.

**Question:** As shown in the figure, if CB = 4.0, DB = 7.0, and D is the midpoint of AC, then the length of AC is ()
A. 6cm      B.7cm      C.4cm      D.5cm

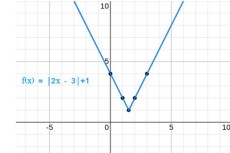

**Switch Frequency = 3.28**

To find the length of AC, we can use the fact that D is the midpoint of AC. This means that AD = DC. Since CB = 4.0 and DB = 7.0, we can find the length of AC by adding CB and DB.
AC = CB + DB = 4.0 + 7.0 = 11.0
However, this is not one of the given options. Let's re-evaluate the problem. Since D is the midpoint of AC, AD = DC. This means that AC = 2 * AD. We can find the length of AD by subtracting CB from DB.
AD = DB - CB = 7.0 - 4.0 = 3.0
Now, AC = 2 * AD = 2 * 3.0 = 6.0
Therefore, the length of AC is 6.0 cm.
**Answer: A**

**Switch Frequency = 3.51**

To find the length of AC, we can use the fact that D is the midpoint of AC. This means that AD = DC. Since CB = 4.0 and DB = 7.0, we can find the length of AC by adding CB and DB together. Therefore, AC = CB + DB = 4.0 + 7.0 = 11.0. However, since D is the midpoint of AC, AD = DC = 11.0 / 2 = 5.5. Therefore, the length of AC is 5.5 cm.

Figure 10: From MathVista, $pidx = 6$. Including results generated under different overall switch frequency.

**Question:** The derivative of f(x) at x=2 is ____ that at x=5
A. larger than      B. equal to      C. smaller than

f(x) = |2x - 3|+1

**Switch Frequency = 2.07**

To find the derivative of f(x) at x=2 and x=5, we need to analyze the graph of the function f(x) = |2x - 3| + 1.
At x=2, the function f(x) = |2x - 3| + 1 is a straight line with a slope of 2. The derivative of a straight line is the slope of the line, which is 2.
At x=5, the function f(x) = |2x - 3| + 1 is a straight line with a slope of 2. The derivative of a straight line is the slope of the line, which is 2.
Therefore, the derivative of f(x) at x=2 is equal to the derivative of f(x) at x=5.
**Answer: B**

**Switch Frequency = 3.21**

To find the derivative of f(x) at x=2 and x=5, we need to first find the derivative of the function f(x) = |2x - 3| + 1.
The derivative of f(x) is:
f'(x) = 2, for x > 3/2
f'(x) = -2, for x < 3/2
Now, let's find the derivative at x=2 and x=5:
At x=2, the derivative is f'(2) = 2.
At x=5, the derivative is f'(5) = -2.
Therefore, the derivative of f(x) at x=2 is larger than the derivative at x=5.
**Answer: A**

Figure 11: From MathVista, $pidx = 23$. Including results generated under different overall switch frequency.

Table 7: Comparison of `S-ATM`'s performance under different attention layer used in the calculation of $\alpha_{att}$. Using `qwen2.5-vl-3b`.

| Attention Layer Selection | MathVerse | | | |
|---|---|---|---|---|
| | All | P-Geo. | S-Geo. | Func. |
| Baseline | 34.9 | 37.5 | 18.5 | 39.0 |
| S-ATM (layer 1) | **37.1** ↑2.2 | 38.6 | **20.2** | **44.7** |
| S-ATM (layer 6) | 35.5 ↑0.6 | **39.3** | 18.5 | 38.0 |
| S-ATM (layer 7) | 35.2 ↑0.3 | 36.9 | 18.5 | 42.1 |

Table 8: Comparison of `S-ATM`'s performance under different $\alpha_0$. Using `qwen2.5-vl-3b`.

| $\alpha_0$ Selection | MathVerse | | | |
|---|---|---|---|---|
| | All | P-Geo. | S-Geo. | Func. |
| 0.0 | 31.1 | 35.3 | 11.8 | 32.1 |
| 0.45 | 32.1 | 37.3 | 11.1 | 30.7 |
| 0.51 | **37.1** | **38.6** | **20.2** | **44.7** |
| 0.55 | 35.4 | 37.0 | 19.5 | 42.0 |
| 1.0 | 34.9 | 37.5 | 18.5 | 39.0 |

### A.7 PERFORMANCE ON SCIENCEQA

In this section, we provide `S-ATM`'s performance on ScienceQA (test set, evaluating 2000 image-containing cases). ScienceQA is a multimodal scientific reasoning benchmark that includes scientific diagrams and OCR-heavy tasks, making it different from MathVerse, MathVista and MMStar. As shown in Tab. 9, we evaluated the performance of `qwen2.5-vl-3b` on ScienceQA observed a consistent performance gain. Together with the results on MathVerse, MathVista, and MMStar, these results demonstrate the generalizability of `S-ATM`.

### A.8 OTHER MODEL'S PERFORMANCE

`Llava-onevision-8b-instruct` is a latest LLaVA series VLM model. In this section, we provide `S-ATM`'s performance when using `llava-onevision-8b-instruct`. As shown in Tab. 10, `S-ATM` still achieves stable improvement on MathVerse. This additional results further demonstrates that our method maintains its generalizability and robustness on different series of models ranging from Qwen series, InternVL series and LLaVA series.

Table 9: `S-ATM`'s performance on ScienceQA (test set, evaluating 2000 image-containing cases), including All, N-Sci.(natural science), S-Sci.(social science) and L-Sci.(language science).

| Method | ScienceQA | | | |
|---|---|---|---|---|
| | All | N-Sci. | S-Sci. | L-Sci. |
| Baseline | 72.7 | 68.1 | 79.6 | 81.8 |
| S-ATM | **73.1** ↑0.4 | **68.6** | **79.6** | **84.1** |

Table 10: `S-ATM`'s performance on MathVerse, using `llava-onevision-8b-instruct`.

| Method | MathVerse | | | |
|---|---|---|---|---|
| | All | P-Geo. | S-Geo. | Func. |
| Baseline | 41.9 | 47.5 | 16.0 | 43.4 |
| S-ATM | **43.1** ↑1.2 | **48.4** | **19.3** | **44.0** |

