# OpenReview forum: "S-ATM: Self-Boosting Visual Reasoning via Adaptive Token Merging"
_ICLR.cc/2026/Conference — Submitted to ICLR 2026_

### Official Review · Reviewer_Bs6K · 2025-11-01

**Soundness:** 4
**Presentation:** 4
**Contribution:** 3
**Rating:** 4
**Confidence:** 4

**Summary:**

This paper introduces S-ATM, a training-free decoding strategy that enhances visual reasoning capabilities in Vision-Language Models (VLMs). The method dynamically merges decoding outputs from two parallel pathways—a visual pathway using the original image input and a textual pathway using a caption generated by the same model. The merging is guided by a weighting factor derived from attention allocation over visual tokens, coupled with a momentum-based smoothing mechanism that stabilizes switches between the pathways.

The method is evaluated on visual reasoning benchmarks like MathVerse, MathVista, and MMStar across a range of model sizes (3B to 14B). S-ATM provides consistent improvements over baseline VLM decoding and outperforms caption-augmentation baselines. The authors also run plug-and-play experiments combining different VLMs and LLMs, showing additional capability fusion without extra training.

**Strengths:**

1. the approach requires no retraining or external supervision. It enhances reasoning using only decoding-time modifications, making it easy to plug into existing VLMs.
2. the method intelligently modulates the contribution of visual and textual pathways using visual attention signals and momentum filtering, improving reasoning coherence without sacrificing visual grounding.
3. the paper provides thorough analysis on why S-ATM works, including identifying “forking tokens” where intervention matters and linking momentum smoothing to decoding stability.

**Weaknesses:**

1. While the paper benchmarks S-ATM primarily on MathVerse, MathVista, and MMStar—datasets that are well-suited for testing visual mathematical reasoning—this narrow focus limits the scope of its demonstrated effectiveness. These datasets emphasize symbolic reasoning, geometry-based perception, or graph comprehension, which, while important, represent only a subset of real-world multimodal reasoning challenges. Broader evaluation across commonsense visual reasoning tasks (e.g., VCR, GQA), diagram-based or scientific illustrations (e.g., AI2D, ScienceQA), and multi-image, temporal, or story-based tasks (e.g., CLEVR, VizWiz, or VideoQA datasets) would better reflect the practical utility and general applicability of S-ATM.
2. The approach relies heavily on self-generated captions. No ablation or analysis is provided on caption quality—e.g. what happens if the model generates low-quality descriptions due to hallucination or cluttered images.
3. The paper does not include any quantitative analysis of the additional computational costs introduced by running two decoding pathways in parallel during inference. Since S-ATM requires simultaneous token generation across both the visual and caption-based textual streams, and further performs per-token attention analysis and momentum-based merging, it likely incurs additional latency and memory overhead relative to baseline decoding. For real-world systems that operate under strict resource or latency constraints (e.g., edge deployment, real-time QA), this omission makes it difficult to assess the practical scalability or deployment trade-offs of S-ATM.
4. S-ATM is primarily applied on Qwen-based VLMs (Qwen2.5-VL and InternVL variants), which, while popular, represent only a subset of modern multimodal architectures. The absence of results on other widely used VLM families—such as LLaVA, Gemma, etc—restricts the generalizability of the findings. Since different VLMs use varying fusion mechanisms, visual encoders, and attention tokenization schemes, it’s unclear whether S-ATM’s adaptive merging and momentum smoothing would transfer as effectively to other architectures.

**Questions:**

1. Does the method help with grounding consistency or reduce hallucination in image-based QA?
2. What happens if visual attention is noisy or misallocated?

---

> ### Author Response · Authors · 2025-11-20
> **Rebuttal by Authors [1/2]**
>
> Thank you for your supportive review and suggestions. Below we respond to the comments in **Weaknesses (W) and Questions (Q).**
>
> ---
>
> > ***W1: Broader evaluation across commonsense visual reasoning tasks, diagram-based or scientific illustrations, and multi-image, temporal, or story-based tasks would better reflect the practical utility and general applicability of S-ATM.***
> >
>
> Firstly, we would like to point out that our core focus is visual reasoning. The math/VQA benchmarks we choose—MathVerse, MathVista, and MMStar—are well aligned with this task, making them reasonable choices. These benchmarks choices are also consistent with those used in a previous work [1].
>
> At the same time, we agree that including additional benchmarks can make our conclusions more convincing. Following your suggestion, we added experiments of S-ATM on **ScienceQA** [2] (test set, evaluating ~2000 image-containing cases), a multimodal scientific benchmark that involves scientific diagrams and OCR-heavy tasks, making it different from the previous math/VQA benchmarks. As shown in ***Table 9*** (page 18), S-ATM again achieves consistent improvements. Together with the earlier results on MathVerse, MathVista, and MMStar, these findings clearly demonstrate the robustness of our method.
>
>  **Reference:**
>
> [1] Chen, Shiqi, et al. "Bring reason to vision: Understanding perception and reasoning through model merging." *arXiv preprint arXiv:2505.05464* (2025).
>
> [2] Saikh T, Ghosal T, Mittal A, et al. Scienceqa: A novel resource for question answering on scholarly articles[J]. International Journal on Digital Libraries, 2022, 23(3): 289-301.
>
> ---
>
> > ***W2: The approach relies heavily on self-generated captions. No ablation or analysis is provided on caption quality.***
> >
>
> Following your suggestion, we add the discussion on the impact of caption quality. Since S-ATM is designed as a self-boosting method, the captions used in ***Table 1*** (page 5) are generated by the model itself. For a small model like Qwen2.5-VL-3B, its caption quality is relatively low. Here, we provide extra comparison experiments on caption quality. We compare self-generated captions with two higher-quality captions: those generated by InternVL3.5-8B, and the text-dominant caption from MathVerse. As shown in ***Table 5*** (page 8), higher-quality captions indeed contribute more to boosting the model’s reasoning ability.
>
> Besides, based on our main results in ***Table 1*** (page 5), we would like to further conclude that: 1) vanilla captioning using strategies like CapIn, CapOut, and CapOnly fails to enhance reasoning, while only S-ATM succeeds; 2) S-ATM can still boost performance even when using lower-quality (self-generated) captions, demonstrating its robustness to boosting visual reasoning ability without external data.

---

> ### Author Response · Authors · 2025-11-20
> **Rebuttal by Authors [2/2]**
>
> > ***W3: The paper does not include any quantitative analysis of the additional computational costs introduced by running two decoding pathways in parallel during inference.***
> >
>
> Following your suggestion, we add extra analysis to time and computation efficiency. S-ATM introduces additional pathway with extra, but acceptable, costs. On average, token merging accounts for only **0.43%** of the total time per token. To enable a fair comparison, we added an experiment where the baseline runs two inferences and record the **pass@2** accuracy. In this setting, the baseline's FLOPs are the same as ours (and even exceed S-ATM), allowing for a more equitable comparison. As shown in ***Table 4*** (page 7), S-ATM still achieves better performance, further demonstrating the effectiveness of our method under stricter comparisons.
>
> During actual implementation, decoding can be parallelized on a single model by setting the batch size to 2, and the additional memory overhead does not double (from **~8 GiB** to **~13 GiB** for a 3B model).
>
> Overall, S-ATM can be understood as a parallel decoding strategy that appropriately scales test-time computation costs to achieve stronger visual reasoning ability. Recent works, such as Kimi-K2-Thinking [3], ParaThinker [4], and Parallel-r1 [5], also further confirm the promising research potential of parallel decoding for future reasoning tasks.
>
> **Reference:**
>
> [3] https://huggingface.co/moonshotai/Kimi-K2-Thinking
>
> [4] Wen H, Su Y, Zhang F, et al. Parathinker: Native parallel thinking as a new paradigm to scale llm test-time compute[J]. arXiv preprint arXiv:2509.04475, 2025.
>
> [5] Zheng T, Zhang H, Yu W, et al. Parallel-r1: Towards parallel thinking via reinforcement learning[J]. arXiv preprint arXiv:2509.07980, 2025.
>
> ---
>
> > ***W4: The absence of results on other widely used VLM families—such as LLaVA, Gemma, etc—restricts the generalizability of the findings.***
> >
>
> Following your suggestion, we conducted experiments on **LLaVA-OneVision-1.5-8B-Instruct** [6], the latest model in LLaVA series.  As shown in ***Table 10*** (page 18), S-ATM continues to achieve stable improvement on the benchmark. This additional results on three different series of models further demonstrates the generalizability and robustness across different model architectures ranging from Qwen series, InternVL series and LLaVA series.
>
> **Reference:**
>
> [6] An X, Xie Y, Yang K, et al. Llava-onevision-1.5: Fully open framework for democratized multimodal training[J]. arXiv preprint arXiv:2509.23661, 2025.
>
> ---
>
> > **Q1:** We hope to point out that the core contribution of S-ATM is to self-boost VLM’s performance on reasoning-based benchmarks. Even so, a detailed discussion on its generalizability to out-of-domain benchmarks (e.g., image-centric/perception heavy benchmarks like **MMStar** and **ScienceQA**), along with experimental results, is provided in **W1**.
> >
>
> > **Q2:** This is just the purpose of our proposed momentum smoothing process — to prevent visual attention’s noise or misallocation from affecting the overall consistency, thereby causing inappropriate pathway selection.
> >

---

### Official Review · Reviewer_Po6T · 2025-11-01

**Soundness:** 2
**Presentation:** 3
**Contribution:** 2
**Rating:** 4
**Confidence:** 3

**Summary:**

The paper proposes S-ATM, a training-free decoding strategy that enhances visual reasoning in VLMs without using external priors. It adaptively merges two decoding pathways—original image–text and self-generated caption–text—guided by visual attention and stabilized via momentum smoothing. Experiments show improved reasoning coherence and stability, highlighting the role of token-level dynamics in long-chain visual reasoning.

**Strengths:**

- This paper introduces a new decoding method for enhancing reasoning capability of VLM

- With extensive experiments, the author demonstrated that S-ATM is effective in terms of accuracy

**Weaknesses:**

- Although I do not find the core idea itself particularly novel, I believe it is contributive and thus acceptable. The experiments also comprehensively evaluate the proposed S-ATM across various models in terms of accuracy, effectively demonstrating its effectiveness. However, since the proposed approach is a decoding method, I think showing its efficiency would be more practically meaningful. Because the method generates two pathways (text and vision) and performs adaptive weighting at the attention level to produce the final logits for decoding, the inference process seems potentially time-consuming. I would like to know how fast the actual inference speed is in practice.

- Although the proposed method achieves higher performance than the baseline, I am curious about its reliability. This is because the approach involves several hyperparameters (e.g., $\alpha_0$,  $\epsilon$. etc), and merging two pathways at each layer may introduce additional variability. If these parameters must be newly adapted or optimized for each benchmark or model, it could raise concerns about the generalizability of the method.

- In Table 1, how was the baseline score measured? Was it obtained by prompting the model to produce a direct answer, or did you employ Chain-of-Thought (CoT) prompting?

**Questions:**

See Weaknesses.

---

> ### Author Response · Authors · 2025-11-20
> **Rebuttal by Authors**
>
> Thank you for your supportive review and suggestions. Below we respond to the comments in **Weaknesses (W)**.
>
> ---
>
> > ***W1:  I think showing its efficiency would be more practically meaningful.  I would like to know how fast the actual inference speed is in practice.***
> >
>
> Following your suggestion, we add extra analysis to time and computation efficiency. S-ATM introduces additional pathway with extra, but acceptable, costs.On average, token merging contributes only **0.43%** to the per-token runtime, and the total number of generated tokens increases by a factor of **×1.26**. To enable a fair comparison, we added an experiment where the baseline runs two inferences and record the **pass@2** accuracy. In this setting, the baseline's FLOPs are the same as ours (and even exceed S-ATM), allowing for a more equitable comparison. As shown in ***Table 4*** (page 7), S-ATM still achieves better performance, further demonstrating the effectiveness of our method under stricter comparisons.
>
> During actual implementation, decoding can be parallelized on a single model by setting the batch size to 2, and the additional memory overhead does not double (from **~8 GiB** to **~13 GiB** for a 3B model).
>
> Overall, S-ATM can be understood as a parallel decoding strategy that appropriately scales test-time computation costs to achieve stronger visual reasoning ability. Recent works, such as Kimi-K2-Thinking [1], ParaThinker [2], and Parallel-r1 [3], also further confirm the promising research potential of parallel decoding for future reasoning tasks.
>
> **Reference:**
>
> [1] https://huggingface.co/moonshotai/Kimi-K2-Thinking
>
> [2] Wen H, Su Y, Zhang F, et al. Parathinker: Native parallel thinking as a new paradigm to scale llm test-time compute[J]. arXiv preprint arXiv:2509.04475, 2025.
>
> [3] Zheng T, Zhang H, Yu W, et al. Parallel-r1: Towards parallel thinking via reinforcement learning[J]. arXiv preprint arXiv:2509.07980, 2025.
>
> ---
>
> > ***W2: The approach involves several hyperparameters and merging two pathways at each layer may introduce additional variability. If these parameters must be newly adapted or optimized for each benchmark or model, it could raise concerns about the generalizability of the method.***
> >
>
> Thank you for the kind suggestions and here we provide a detailed clarification regarding hyperparameter selection. As mentioned in ***L109*** and ***Eq. (2)*** of the paper, we perform adaptive merging only on the final-layer logits. In total, the parameters that need to be adjusted during experiments are **$\epsilon$**, **$\gamma$**, and $\alpha_0$. Among them, $\epsilon$ and $\gamma$ are discussed in detail in ***Section 5.4*** (page 9) of the paper. In practice, we typically fix them to **0.01** and **0.9**, respectively.
>
> The parameter $\alpha_0$ needs to be adjusted across benchmarks. We conducted an ablation study for it on MathVerse, as shown in ***Table 8*** (page 17). As can be seen, **0.51** is the optimal value we selected for MathVerse. In practice, we only perform a small-range search on the validation set for at most the five values **[0.5, 0.51, 0.53, 0.55, 0.57].** Thus, the search space is not large. The cost of adapting our method to new benchmarks is therefore low.
>
> ---
>
> > ***W3: In Table 1, how was the baseline score measured? Was it obtained by prompting the model to produce a direct answer, or did you employ Chain-of-Thought (CoT) prompting?***
> >
>
> Following your advice, we provide a detailed clarification of our evaluation process. The benchmarks we use require detailed reasoning before producing an answer, so we prompt the model do “necessary step-by-step reasoning before giving the final answer.” After that, we follow the official evaluation pipelines of each benchmark: we use an LLM to extract the predicted option from the generated text, compare it with the ground truth, and compute the accuracy.

---

### Official Review · Reviewer_DNrB · 2025-11-01

**Soundness:** 2
**Presentation:** 3
**Contribution:** 3
**Rating:** 6
**Confidence:** 4

**Summary:**

This paper introduces a training-free, model-agnostic decoding strategy for vision–language models. The method runs two parallel reasoning paths—one conditioned on the image and one that relies on a self-produced textual description—and fuses their token predictions with a lightweight, step-wise gate. The goal is to exploit complementary strengths of image-grounded and language-only reasoning at inference time, requiring no finetuning or architectural changes. Experiments across a range of multimodal benchmarks report modest but consistent improvements, with ablations indicating that both the caption detour and the gating/smoothing heuristics contribute to the gains.

**Strengths:**

- The idea is novel to me, and is easy to follow. It only requires two parallel decoding paths with an attention-derived gate plus a one-state momentum smoother. The pipeline is straightforward.
- Results on MathVerse, MathVista, and MMStar show modest but steady improvements.
- The paper links improvements to high-entropy “forking” tokens and argues that smoothing reduces excessive pathway switching, offering a plausible account of when and why the method helps. It's interesting.

**Weaknesses:**

> Technical

1. I think take attention as a gating signal is brittle.
   - The image-attention ratio \\(\alpha_{\text{att}}\\) is taken from a chosen layer, averaged over heads/tokens, and then mapped linearly to \\(\delta_t\\) with a hard heuristic. Which layer/heads are used is not specified, no evidence is provided that attention mass correlates with the *causal necessity* of vision at step \\(t\\). Attention is a weak proxy for importance.

2. Momentum analysis rests on unrealistic assumptions.
  -  The "proof" that momentum reduces switching frequency assumes i.i.d. Gaussian \\(\alpha_t\\), then studies sign flips; actual \\(\alpha_t\\) is bounded and derived from attention, and the smoothed series is not driven by i.i.d. noise. The derivation does not establish task accuracy improvements, only a toy decrease in flips.

3.  S-ATM runs two full decoders in parallel and generates a long caption first . There is no wall-clock analysis or FLOPs comparison, nor a cost-normalized metric (e.g., accuracy per second).

> Experiments

4. No study of:
  - which layer to read attention from
  - effect of temperature and caption length
  - probability vs logit-space fusion

5. Benchmarks are mostly math/VQA. No tests on instruction-following multimodal datasets, long-context OCR-heavy tasks, or robustness.

**Questions:**

1. Why logit mixing? Please justify Eq. (5) versus (i) probability-space mixture \\(p=\alpha p_{\text{vis}}+(1-\alpha)p_{\text{res}}\\) and (ii) log-linear/geometric fusion \\(\log p \propto \alpha\log p_{\text{vis}}+(1-\alpha)\log p_{\text{res}}\\). Any calibration of logit scales between pathways?

2. You say "a specific layer" provides \\(A\\) (Eq. (3)). Which layer and why? How sensitive is S-ATM to this choice and to head selection? Any results averaging multiple layers or using value/FFN activations?

3. The text states "values <0.1 are mapped to \\([-\varepsilon,\varepsilon]\\)" to obtain \\(\delta_t\\) (Eq. (4)). Please provide the exact formula, bounds on \\(\alpha_t\\), and an ablation over \\(\alpha_0,\varepsilon\\).

4. Can you empirically link “fewer sign flips” to accuracy improvements while holding other factors constant? Also, replace the i.i.d. Gaussian assumption with a model consistent with your measured \\(\alpha_t\\) statistics.

5. What is the end-to-end latency and GPU memory overhead of S-ATM vs baseline for the same max tokens? Include captioning time

6. How are forking tokens detected at test time without labels? Your effective-token definition depends on pathway disagreement; does an oracle-free detector exist if one wanted to apply S-ATM only at those positions?

7. Did you try a learned scalar gate for \\(\alpha_t\\) (frozen VLM with tiny regressor over hidden states) or confidence-aware gating (entropy/variance of each pathway)?

8. If attention extraction is unavailable, can \\(\alpha_t\\) be computed from pathway self-statistics and how does that compare?

9. You merge *logits* linearly as \\(z_t = \\alpha_t z^{\\text{vis}}_t + (1-\\alpha_t) z^{\\text{res}}_t\\) . Why is logit-space interpolation appropriate given that logits are only defined up to affine transforms and may be differently calibrated across the two pathways? Did you compare against probability-space mixture \\(p = \\alpha_t \\operatorname{softmax}(z^{\\text{vis}}_t) + (1-\\alpha_t) \\operatorname{softmax}(z^{\\text{res}}_t)\\) or *log-linear* fusion \\(\\log p \\propto \\alpha_t \\log p^{\\text{vis}} + (1-\\alpha_t) \\log p^{\\text{res}}\\)? Any temperature or scale calibration to align the two streams before fusion?

10. The “theory” models \\(\\alpha_t\\) as i.i.d. \\(\\mathcal{N}(0,\\sigma^2)\\) and studies sign flips after momentum. But \\(\\alpha_t\\) is derived deterministically from bounded attention statistics and is highly autocorrelated. Can you re-derive the result under a bounded, autocorrelated process?

11. Beyond reducing “switch frequency,” is there measurable improvement in probability calibration or search stability (e.g., lower variance of token log-likelihoods across beams)? Any curves showing accuracy vs. switch-frequency trade-off while sweeping \\(\\gamma\\) and \\(\\varepsilon\\)?

12. You use temperature 0.1 throughout. How do outcomes change with higher temperatures or nucleus/top-k sampling? Does higher randomness amplify mis-calibration between branches and thus favor probability-space or log-linear fusion over logit mixing?

13. Some cells show small drops (e.g., InternVL3.5-14B on certain MathVista partitions). Can you quantify when S-ATM hurts—e.g., perception-dominated items where visual grounding is essential? A per-category error breakdown would help.

14. Running two full decoders plus caption generation likely doubles latency and memory. What is the end-to-end wall-clock and GPU RAM overhead vs. baseline for the same max tokens?

15. When is forward guidance harmful? Consider a case where attention briefly dips on image tokens during visually crucial steps (e.g., small visual cue). The mapping that pushes \\(\\alpha_t\\) toward the textual pathway might wrongly suppress vision. Any safeguards (e.g., hard floor on \\(\\alpha_t\\) when visual entropy is high)?








Overall, I like this work. However, I still have many concerns regarding the technical issues and experiments. I will adjust the rating based on the authors' feedback/rebuttal.

---

> ### Author Response · Authors · 2025-11-20
> **Rebuttal by Authors [1/3]**
>
> Thank you for your supportive review and suggestions. Below we respond to the comments in **Weaknesses (W) and Questions (Q).**
>
> ---
>
> > ***W1: I think take attention as a gating signal is brittle. Which layer/heads are used is not specified, no evidence is provided that attention mass correlates with the causal necessity of vision at step.***
> >
>
> Following your suggestions, we specify here that our experiments use the **first layer‘s attention map**, since shallow attention layers in VLMs typically align low-level visual features with textual tokens (perception + grounding), which fits our setting better. Ablation study on attention layer selection are shown in ***Table7*** (page 17).
>
> In addition, from previous papers, we also identified research that uses attention to reflect perception importance or analyze how image-token attention ratio affects overall performance.
>
> - **More Thinking, Less Seeing** [1]: discusses how attention allocation influences the relationship between reasoning and perception during the decoding process.
> - **Interleaved-Modal Chain-of-Thought** [2]: uses image-token attention to reinsert important image patches at specific points during decoding.
> - **MLLMs Know Where to Look** [3]: studies the correlation between image-token attention ratios and key image regions during decoding.
>
> These works further demonstrate that attention signals reflect the importance of perception during decoding. We hope that the above discussion can clearly convey the motivation and methodology behind our design.
>
> **Reference:**
>
> [1] Liu C, Xu Z, Wei Q, et al. More Thinking, Less Seeing? Assessing Amplified Hallucination in Multimodal Reasoning Models[J]. arXiv preprint arXiv:2505.21523, 2025.
>
> [2] Gao J, Li Y, Cao Z, et al. Interleaved-modal chain-of-thought[C]//Proceedings of the Computer Vision and Pattern Recognition Conference. 2025: 19520-19529.
>
> [3] Zhang J, Khayatkhoei M, Chhikara P, et al. Mllms know where to look: Training-free perception of small visual details with multimodal llms[J]. arXiv preprint arXiv:2502.17422, 2025.
>
> ---
>
> > ***W2: Momentum analysis rests on unrealistic assumptions.***
> >
>
> Thank you for the observation. In ***Section 3.3*** (page 4), the analysis simplifies the switch frequency of a continuous sequence of N tokens into N independent Gaussian variables. While this indeed cannot fully provide a quantitative explanation of our method, our main purpose is to offer a mathematical illustration of the momentum smoothing process — aiming to convey, in a semi-qualitative and semi-quantitative manner, a clearer intuition for the motivation and objective of momentum smoothing.
>
> The quantitative analysis is presented in ***Section 5.4*** (page 9) and ***Figure 6*** (page 8), where we show and analyze that the strength of momentum smoothing is positively correlated with the stability during decoding. ***Section 5.4*** (page 9) also discuss the impact of momentum smoothing on overall accuracy.
>
> ---
>
> > ***W3: S-ATM runs two full decoders in parallel and generates a long caption first. There is no wall-clock analysis or FLOPs comparison, nor a cost-normalized metric.***
> >
>
> Following your suggestion, we add extra analysis to time and computation efficiency. S-ATM introduces additional pathway with extra, but acceptable, costs. On average, token merging accounts for only **0.43%** of the total time per token. To enable a fair comparison, we added an experiment where the baseline runs two inferences and record the **pass@2** accuracy. In this setting, the baseline's FLOPs are the same as ours (and even exceed S-ATM), allowing for a more equitable comparison. As shown in ***Table 4*** (page 7), S-ATM still achieves better performance, further demonstrating the effectiveness of our method under stricter comparisons.
>
> During actual implementation, decoding can be parallelized on a single model by setting the batch size to 2, and the additional memory overhead does not double (from **~8 GiB** to **~13 GiB** for a 3B model).
>
> Overall, S-ATM can be understood as a parallel decoding strategy that appropriately scales test-time computation costs to achieve stronger visual reasoning ability. Recent works, such as Kimi-K2-Thinking [4], ParaThinker [5], and Parallel-r1 [6], also further confirm the promising research potential of parallel decoding for future reasoning tasks.
>
> **Reference:**
>
> [4] https://huggingface.co/moonshotai/Kimi-K2-Thinking
>
> [5] Wen H, Su Y, Zhang F, et al. Parathinker: Native parallel thinking as a new paradigm to scale llm test-time compute[J]. arXiv preprint arXiv:2509.04475, 2025.
>
> [6] Zheng T, Zhang H, Yu W, et al. Parallel-r1: Towards parallel thinking via reinforcement learning[J]. arXiv preprint arXiv:2509.07980, 2025.

---

> ### Author Response · Authors · 2025-11-20
> **Rebuttal by Authors [2/3]**
>
> > ***W4: No study of: 1 which layer to read attention from 2 effect of temperature and caption 3 probability vs logit-space fusion***
> >
>
> We appreciate your kind suggestions. And here we add the necessary ablations in the revised version:
>
> - **For the attention-layer ablation:** the results are shown in ***Table 7*** (page 17). Detailed discussion are provided in W1.
> - **For temperature:** We set the temperature to 0.1 to ensure **stability** and **fairness** in the experimental results, based on three repeated runs on MathVerse showing only 0.1% variation. Besides, VLM decoding papers (e.g. Look Twice Before You Answer [7]) tend to choose a low temperature, as it is a reasonable and fair choice for such tasks. Thus, we fixed a temperature of 0.1 across different benchmarks and models for fair comparison.
> - **For caption quality:** the results are shown in ***Table 5***  (page 8). We compared self-generated captions with two higher-quality captions: those generated by InternVL3.5-8B and MathVerse’s text-dominant captions. The results show that higher-quality captions indeed help boost the model’s reasoning ability. Overall, S-ATM already works well in the self-boosting setting without external information, and becomes even better when high-quality captions are available.
> - **For probability vs. logit-space fusion:** As mentioned in ***L109*** and ***Eq. (2)*** of the paper, the logit and probability spaces are essentially the same thing (the latter simply applies a softmax layer to the former). If your interpretation of logits differs from ours, please kindly specify it—we are willing to provide additional results.
>
> **Reference:**
>
> [7] Zou, Xin, et al. "Look twice before you answer: Memory-space visual retracing for hallucination mitigation in multimodal large language models." *arXiv preprint arXiv:2410.03577* (2024).
>
> ---
>
> > ***W5:  Benchmarks are mostly math/VQA. No tests on instruction-following multimodal datasets, long-context OCR-heavy tasks, or robustness.***
> >
>
> Firstly, we would like to point out that our core focus is visual reasoning. The math/VQA benchmarks we choose—MathVerse, MathVista, and MMStar—are well aligned with this task, making them reasonable choices. These benchmarks choices are also consistent with those used in a previous work [8].
>
> At the same time, we agree that including additional benchmarks can make our conclusions more convincing. Following your suggestion, we added experiments of S-ATM on **ScienceQA** [9] (test set, evaluating ~2000 image-containing cases), a multimodal scientific benchmark that involves scientific diagrams and OCR-heavy tasks, making it different from the previous math/VQA benchmarks. As shown in ***Table 9*** (page 18), S-ATM again achieves consistent improvements. Together with the earlier results on MathVerse, MathVista, and MMStar, these findings clearly demonstrate the robustness of our method.
>
>  **Reference:**
>
> [8] Chen, Shiqi, et al. "Bring reason to vision: Understanding perception and reasoning through model merging." *arXiv preprint arXiv:2505.05464* (2025).
>
> [9] Saikh T, Ghosal T, Mittal A, et al. Scienceqa: A novel resource for question answering on scholarly articles[J]. International Journal on Digital Libraries, 2022, 23(3): 289-301.

---

> ### Author Response · Authors · 2025-11-20
> **Rebuttal by Authors [3/3]**
>
> > **Q1, Q9:** Detailed definition of logits is provided in **W4**. Besides, log-linear/geometric fusion is also a possible strategy. The reason we did not adopt it is that we wanted to preserve the model's original probability distribution properties (e.g., entropy), but it remains a possible strategy for future work. Thank you for your suggestion.
> >
>
> > **Q2:** Detailed discussion on attention layer selection are provided in **W1**.
> >
>
> > **Q3:** The exact formula is: $\delta_t = - \epsilon + 2\epsilon*\alpha_{att}$.
> >
>
> > **Q4, Q10, Q11:** Detailed analysis on switch frequency / sign flips are provided in ***Section 5.4*** (page 9); Since precisely fitting $\alpha_t$ into an analytic function would be highly complex, we instead adopt a Gaussian distribution as a reasonable simplified model based on our empirical observations. Precise quantitative analysis will be conducted in future work if needed.
> >
>
> > **Q5, Q14:** Detailed discussion on computation and memory costs are provided in **W3**.
> >
>
> > **Q6:** Forking tokens are identified as those with token probability distributions in the top 20% of entropy (its official definition). We use this to detect them; An oracle-free detector is a promising idea, though it may require extensive labeled data and may vary across different VLMs. Still, it offers a potential way to reduce computation cost and could be a direction for future work. Thank you for your suggestion.
> >
>
> > **Q7:** We aimed to develop a training-free method, so we did not attempt other methods that require training; Entropy is a good symbol of effective tokens but it lacks evidence showing its relationship with perception importance compared with our proposed attention driven method.
> >
>
> > **Q8:** Under this assumption, we could consider learning a scalar gate, as you mentioned.
> >
>
> > **Q12:** Detailed discussion on temperature are provided in **W4**. We aim to explain that fixing the temperature to 0.1 is a reasonable, fair, and general approach in the VLM decoding field.
> >
>
> > **Q13:** Based on our observation, some cases in MathVista almost entirely rely on perception rather than reasoning (such as some geometry and chart-based questions). In these cases, incomplete information provided by the captions may affect the quality of the textual pathway’s logits, leading to small drops in some circumstances. Nevertheless, we think that the other settings and the newly added results in ***Table 9,10*** (page 18) are sufficient to show the robustness of S-ATM.
> >
>
> > **Q15:** This is just the purpose of our proposed momentum smoothing process — to prevent small clues from affecting the overall consistency, thereby causing inappropriate pathway selection.
> >

---

### Official Review · Reviewer_CAKB · 2025-11-04

**Soundness:** 2
**Presentation:** 2
**Contribution:** 2
**Rating:** 4
**Confidence:** 3

**Summary:**

This paper proposed S-ATM, a training-free decoding strategy for VLMs. Across MathVerse, MathVista, and MMStar, S-ATM yields consistent but modest gains (typically ~+0.5–3%, up to ~+5% on some MathVerse subsets) and can further improve via a plug-and-play variant that uses an external LLM on the textual path. Analyses suggest S-ATM mostly “kicks in” at high-entropy forking tokens—decision points where the two paths disagree—and that smoothing reduces harmful path-switching during long-chain reasoning.

**Strengths:**

1. Simple, training-free mechanism that is easy to bolt onto existing VLMs.

2. The paper has a broad evaluation on three benchmarks and multiple model sizes shows generally positive gains, with larger boosts on math-reasoning subsets.

**Weaknesses:**

1. Decoding cost roughly doubles at inference (two parallel passes + attention inspection each step), but runtime/latency overhead isn’t reported. Thus I feel the gains of the approach are modest and limited.

2. Momentum stability is motivated by a simplified IID Gaussian sign-flip analysis; it’s insightful but far from the true non-Gaussian, autoregressive dynamics of modern decoders.

3. Benchmarks skew toward visual math/VQA; generalization to OCR-heavy, chart/diagram QA, video VQA, or real-world multi-image tasks remains untested. For example, you can conduct experiments on Video VQA benchmarks such Video-MME.

4. Plug-and-play improvements rely on stronger LLMs for the textual path. We don't know if this gain comes from powerful baseline.

**Questions:**

See weaknesses for more details.

---

> ### Author Response · Authors · 2025-11-20
> **Rebuttal by Authors [1/2]**
>
> Thank you for your supportive review and suggestions. Below we respond to the comments in **Weaknesses (W)**.
>
> ---
>
> > ***W1: Decoding cost roughly doubles at inference (two parallel passes + attention inspection each step), but runtime/latency overhead isn’t reported.***
> >
>
> Following your suggestion, we add extra analysis to time and computation efficiency. S-ATM introduces additional pathway with extra, but acceptable, costs. On average, token merging accounts for only **0.43%** of the total time per token, and the total number of generated tokens increases by a factor of **×1.26**. To enable a fair comparison, we added an experiment where the baseline runs two inferences and record the **pass@2** accuracy. In this setting, the baseline's FLOPs are the same as ours (and even exceed S-ATM), allowing for a more equitable comparison. As shown in ***Table 4*** (page 7), S-ATM still achieves better performance, further demonstrating the effectiveness of our method under stricter comparisons.
>
> During actual implementation, decoding can be parallelized on a single model by setting the batch size to 2, and the additional memory overhead does not double (from **~8 GiB** to **~13 GiB** for a 3B model).
>
> Overall, S-ATM can be understood as a parallel decoding strategy that appropriately scales test-time computation costs to achieve stronger visual reasoning ability. Recent works, such as Kimi-K2-Thinking [1], ParaThinker [2], and Parallel-r1 [3], also further confirm the promising research potential of parallel decoding for future reasoning tasks.
>
> **Reference:**
>
> [1] https://huggingface.co/moonshotai/Kimi-K2-Thinking
>
> [2] Wen H, Su Y, Zhang F, et al. Parathinker: Native parallel thinking as a new paradigm to scale llm test-time compute[J]. arXiv preprint arXiv:2509.04475, 2025.
>
> [3] Zheng T, Zhang H, Yu W, et al. Parallel-r1: Towards parallel thinking via reinforcement learning[J]. arXiv preprint arXiv:2509.07980, 2025.
>
> ---
>
> > ***W2: Momentum stability is motivated by a simplified IID Gaussian sign-flip analysis; it’s insightful but far from the true non-Gaussian, autoregressive dynamics of modern decoders.***
> >
>
> Thank you for the observation. In ***Section 3.3*** (page 4), the analysis simplifies the switch frequency of a continuous sequence of N tokens into N independent Gaussian variables. While this indeed cannot fully provide a quantitative explanation of our method, our main purpose is to offer a mathematical illustration of the momentum smoothing process — aiming to convey, in a semi-qualitative and semi-quantitative manner, a clearer intuition for the motivation and objective of momentum smoothing.
>
> The quantitative analysis is presented in ***Section 5.4*** (page 9) and ***Figure 6*** (page 8), where we show and analyze that the strength of momentum smoothing is positively correlated with the stability during decoding. ***Section 5.4*** (page 9) also discuss the impact of momentum smoothing on overall accuracy.

---

> ### Author Response · Authors · 2025-11-20
> **Rebuttal by Authors [2/2]**
>
> > ***W3: Benchmarks skew toward visual math/VQA; generalization to OCR-heavy, chart/diagram QA, video VQA, or real-world multi-image tasks remains untested.***
> >
>
> Firstly, we hope like to point out that our core focus is visual reasoning. The math/VQA benchmarks we choose—MathVerse, MathVista, and MMStar—are well aligned with this task, making them reasonable choices. These benchmarks choices are also consistent with those used in a previous work [4].
>
> At the same time, we agree that including additional benchmarks can make our conclusions more convincing. Following your suggestion, we added experiments of S-ATM on **ScienceQA** [5] (test set, evaluating ~2000 image-containing cases), a multimodal scientific benchmark that involves scientific diagrams and OCR-heavy tasks, making it different from the previous math/VQA benchmarks. As shown in ***Table 9*** (page 18), S-ATM again achieves consistent improvements. Together with the earlier results on MathVerse, MathVista, and MMStar, these findings clearly demonstrate the generalizability of our method.
>
>  **Reference:**
>
> [4]Chen, Shiqi, et al. "Bring reason to vision: Understanding perception and reasoning through model merging." *arXiv preprint arXiv:2505.05464* (2025).
>
> [5] Saikh T, Ghosal T, Mittal A, et al. Scienceqa: A novel resource for question answering on scholarly articles[J]. International Journal on Digital Libraries, 2022, 23(3): 289-301.
>
> ---
>
> > ***W4: Plug-and-play improvements rely on stronger LLMs for the textual path. We don't know if this gain comes from powerful baseline.***
> >
>
> Thank you for the suggestion.  We have updated the caption of ***Table 3*** (page 6) to make the baselines more prominent.
>
> In ***Table 3*** (page 6), we provide the corresponding baselines. The Textual Pathway CapOnly section in the table shows the baseline results for the stronger LLMs. Since LLMs cannot process images directly, we feed them the captions generated by VLMs of the same scale. Specifically, the baseline for Qwen2.5-3B is **37.9%**, while S-ATM improves it to **38.2%**. For Qwen2.5-7B, the baseline is **36.6%**, and S-ATM increases it to **39.3%**.

---

### Author Response · Authors · 2025-11-20
**Summary of Paper Revision**

Dear Reviewers,

Thank you for the constructive feedback, which are really helpful for us. We have posted responses to the proposed concerns and included additional experiment results.

If there are further comments, we will try our best to provide additional clarification or information.

Best,

The Authors

---

We have also uploaded a **Paper Revision** including additional results and analysis:

- **Section 4.1** (page 5): More information about the implementation detail of S-ATM
- **Section 4.2** (page 6): Extra analysis on the additional time cost and computation cost introduced by S-ATM.
- **Table 3** (page 6): Refined version of the Plug-and-Play performance table. (The baseline performance of the stronger LLM is indicated.)
- **Table 4** (page 7): Accuracy of S-ATM and the baseline’s pass@2 results on MathVerse.
- **Table 5** (page 8): Ablation study and discussion for caption quality
- **Table 7** (page 17): Ablation study and hyperparameter selection for attention layers
- **Table 8** (page 17): Ablation study and hyperparameter selection for $\alpha_0$
- **Table 9** (page 18):  Accuracy of S-ATM and baseline on a new benchmark, ScienceQA.
- **Table 10** (page 18): Accuracy of S-ATM and baseline on a newly released model, LLaVA-OneVision-8B-Instruct.

---

### Author Response · Authors · 2025-11-28
**Looking forward to further feedback**

Dear Reviewers,

Thank you again for your valuable comments and suggestions, which are really helpful for us. We have posted responses to the proposed concerns and included additional experiment results.

We totally understand that this is quite a busy period, so we deeply appreciate it if you could take some time to return further feedback on whether our responses solve your concerns. If there are any other comments, we will try our best to address them.

Best,

The Authors

---

### Meta-Review · Area_Chair_QRot · 2025-12-27

**Summary:**

The paper proposes S-ATM, a training-free decoding strategy aimed at mitigating reasoning degradation in Large Vision-Language Models (LVLMs) when processing visual inputs. S-ATM constructs two parallel decoding pathways—one using the original image-text input and the other using a self-generated caption-text input—and adaptively merges their distributions at each step, guided by the model’s attention to visual tokens. A momentum-based smoothing mechanism further stabilizes merging to maintain reasoning coherence. The authors validate S-ATM through experiments on diverse visual reasoning benchmarks, showing it activates at high-entropy reasoning transition tokens and reduces decoding instability.

Reviewers’ core concerns informing the decision included:
* Practical efficiency justification (Reviewer CAKB, Po6T, Bs6K, DNrB);
* Limited empirical experiments (Reviewer CAKB, DNrB);
* Limited novelty (Reviewer Po6T);
* The absence of results on other widely used VLM families (Reviewer Bs6K).

Overall, reviewers have negative scores before the discussion, and the core concerns still stand following the rebuttal. Therefore, the reviewing panel is inclined to recommend a major revision of the paper before it can be considered for acceptance.

**Reviewer Concerns:**

The concerns on the practical efficiency justification for both the proposed method and baselines (Reviewer CAKB, Po6T, Bs6K, DNrB), and the limited empirical experiments (Reviewer CAKB, DNrB) are still outstanding post-rebuttal.

**Reviewer Scores:**

Given that each review has certain concerns that are not fully addressed by the authors’ response, the chance for them to increase their scores is relatively low.

---

### Decision · Program_Chairs · 2026-01-26

Reject